# P113 is a merozoite surface protein that binds the N terminus of *Plasmodium falciparum* RH5

Francis Galaway[1], Laura G. Drought[2], Maria Fala[1], Nadia Cross[2], Alison C. Kemp[2], Julian C. Rayner[2] & Gavin J. Wright[1,2]

Invasion of erythrocytes by *Plasmodium falciparum* merozoites is necessary for malaria pathogenesis and is therefore a primary target for vaccine development. RH5 is a leading subunit vaccine candidate because anti-RH5 antibodies inhibit parasite growth and the interaction with its erythrocyte receptor basigin is essential for invasion. RH5 is secreted, complexes with other parasite proteins including CyRPA and RIPR, and contains a conserved N-terminal region (RH5Nt) of unknown function that is cleaved from the native protein. Here, we identify P113 as a merozoite surface protein that directly interacts with RH5Nt. Using recombinant proteins and a sensitive protein interaction assay, we establish the binding interdependencies of all the other known RH5 complex components and conclude that the RH5Nt-P113 interaction provides a releasable mechanism for anchoring RH5 to the merozoite surface. We exploit these findings to design a chemically synthesized peptide corresponding to RH5Nt, which could contribute to a cost-effective malaria vaccine.

[1] Cell Surface Signalling Laboratory, Wellcome Trust Sanger Institute, Cambridge CB10 1SA, UK. [2] Malaria Programme, Wellcome Trust Sanger Institute, Cambridge CB10 1SA, UK. Correspondence and requests for materials should be addressed to G.J.W. (email: gw2@sanger.ac.uk).

Malaria is a deadly infectious disease caused by *Plasmodium* parasites which is responsible for up to one million deaths annually, primarily in young children living in sub-Saharan Africa[1]. Malaria symptoms result from the blood stages of *P. falciparum* infections when a form of the parasite called the merozoite recognizes and invades host erythrocytes where it replicates asexually[2]. Since invasion is an essential and extracellular step in the parasite lifecycle, it may be targeted by vaccine-induced antibodies[3]. After initial recognition of the host erythrocyte, the pear-shaped merozoite orientates itself so that its apical protuberance is in direct apposition to the host membrane. This triggers the subsequent release of parasite invasion ligands from intracellular secretory organelles including the micronemes and rhoptries[3,4]. An electron-dense nexus between the host and parasite membranes is formed which opens out into a ring-like moving junction which envelops the merozoite, finally resealing behind it, such that the parasite is completely internalized within an intraerythrocytic parasitophorous vacuole[5]. The whole process is rapid, taking just a few seconds[6]. The biochemical interactions involved in invasion are being identified, and their roles in each of these steps determined[4]. Of particular current interest is the interaction between the parasite reticulocyte-binding protein homologue 5 (RH5) and its erythrocyte receptor, basigin[7].

RH5 was first identified by searching the *P. falciparum* genome sequence for homology with the sequences of other RH family members, and the inability to select *RH5*-deficient parasites *in vitro* suggested it was required for blood-stage growth[8]. The role of RH5 as an invasion ligand was established by the identification of basigin as its erythrocyte receptor, and the demonstration that the RH5-basigin interaction was both essential and universally required for invasion[9]. RH5 is detected within the rhoptries of merozoites, relocating to the moving junction during invasion[8]. Live imaging in the presence of fluorescent calcium-sensitive dyes and RH5-basigin interaction antagonists revealed that merozoites could still adhere and deform erythrocytes leading to the conclusion that the RH5-basigin interaction was necessary for, and directly preceded, rhoptry release just before the formation of the moving junction[4]. The protein sequence of RH5 is conserved between strains[10], can elicit antibodies that inhibit parasite growth *in vitro*[11–14], and RH5-based vaccines are protective in a non-human primate *P. falciparum* infection model[15]. These properties of RH5 have made a deeper understanding of its mechanism of action a priority but many basic questions remain unanswered. For example, the lack of any obvious protein sequence feature for anchoring RH5 to a membrane suggests the existence of another mechanism for tethering RH5 to the merozoite surface. In addition, RH5 is detected in parasite culture supernatants as both full length (RH5FL, $\sim 63$ kDa) and processed (RH5Ct, $\sim 45$ kDa) forms but the function of this processing is unknown[8]. Peptide sequencing of purified recombinant RH5 and anti-RH5 antibodies with known epitope locations revealed that RH5Ct lacks the N-terminal region (RH5Nt), which is predicted to be disordered[8,16–18]. RH5Ct folds into a 'kite'-like shape[19,20] and contains a small ($\sim 1,500$ Å$^2$) binding interface for basigin, consistent with the low interaction affinity ($K_D \sim 1 \mu$M)[9], leaving much of its solvent-exposed surface available for simultaneous binding with other proteins. Size separation of parasite proteins from culture supernatants by gel filtration revealed that RH5 is part of a larger ($\sim 200$ kDa) complex, that includes two other parasite proteins: RIPR (RH5 interacting protein)[16] and CyRPA (cysteine-rich protective antigen)[21]. Both RIPR and CyRPA are predicted to be secreted proteins[16,22] (although see ref. 21) which suggests that they do not tether RH5 to the merozoite surface. Because these proteins were identified from bulk parasite extracts

as purified complexes, we do not know whether RIPR and CyRPA bind RH5 directly, nor whether they interact in a mutually exclusive fashion. One further intriguing and unanswered question in relation to the function of RH5 is the role of its N terminus (RH5Nt), which is absent from the processed form of RH5 detected in parasite culture supernatants. This region is predicted to be highly disordered and flexible, to such a degree that it was purposefully removed to promote crystal formation for structural studies[19,20]. Despite this, RH5Nt is particularly well conserved with just a single non-synonymous polymorphism described at very low frequency ($<1\%$) in African isolates[13], and contains the epitope for a potent invasion-blocking monoclonal antibody[18] demonstrating RH5Nt has an important role in RH5 function.

Here we use a sensitive assay that detects interactions between recombinant proteins to show that the glycosylphosphatidylinositol (GPI)-linked protein P113 interacts directly with the N terminus of unprocessed RH5, providing a mechanism by which the RH5 invasion complex is tethered to the merozoite surface. We additionally determine the binding interdependencies between the individual components of the RH5-basigin invasion complex, which, together with RH5 processing, suggests the existence of a one-way switching mechanism that releases the RH5 complex from the merozoite surface during erythrocyte invasion by *P. falciparum*.

## Results

**The N terminus of RH5 has no role in basigin binding**. The *Plasmodium falciparum* RH5 protein is detected as full length and processed forms in both parasite culture supernatants and when expressed recombinantly in either mammalian[13] or insect cells[20]. To identify the sites of processing when expressed in mammalian cells, RH5 was purified, resolved as four bands by SDS–PAGE, and the N terminus of each determined by Edman protein sequencing. The major band (RH5Ct) was consistent with the main processed form of RH5 observed in parasite supernatants (Fig. 1a) and its N terminus is close (14 amino acids C-terminal) to the cleavage site observed when RH5 is expressed in insect cells[20]. The largest band matched the expected mass of the full-length unprocessed protein (RH5FL) and this was confirmed by protein sequencing (Fig. 1a). To determine which of the processed forms were able to interact with the basigin receptor, we made recombinant forms of RH5 corresponding to each of these fragments and tested their ability to bind basigin using a protein interaction assay designed to detect low-affinity extracellular protein interactions called AVEXIS (for AVidity-based EXtracellular protein Interaction Screen[23]) (Fig. 1b). Consistent with the co-crystal structure of RH5 with its receptor, we observed that RH5Ct bound basigin, whereas RH5Nt did not (Fig. 1b). We used surface plasmon resonance (SPR) to show that RH5Ct bound with similar binding parameters to the full-length RH5 protein[9] (Supplementary Table 1; Fig. 1c). Altogether, these data demonstrate that the N-terminal region of RH5 does not play a direct role in basigin binding.

**P113 interacts with the N terminus of RH5**. We reasoned that the cleaved N-terminal region of RH5 might form a distinct binding domain, and because antibodies to this region have shown potent parasite inhibitory activity[18], could be important for RH5 function. One possible role would be to tether RH5 to the surface of the merozoite by interacting with another membrane-anchored parasite protein. We therefore used RH5Nt expressed as a highly avid, pentamerized, enzyme-tagged 'prey' and the AVEXIS assay to systematically screen an existing library

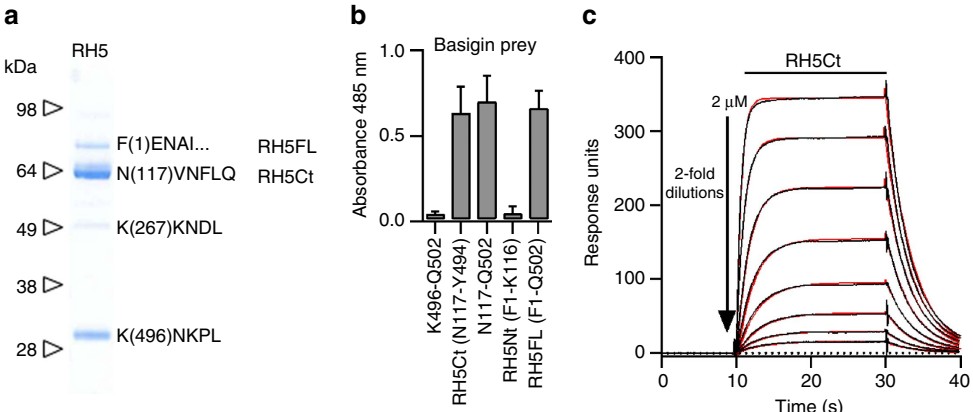

**Figure 1 | The N-terminal domain of RH5 is not involved in basigin binding.** (**a**) Purified tagged recombinant RH5 expressed by HEK293 cells resolved as four species by SDS–PAGE under reducing conditions. The N-terminal sequence of each species was determined by Edman degradation and is shown. (**b**) The N-terminal region of RH5 is not involved in basigin binding. The indicated RH5 fragments were expressed as biotinylated bait proteins and probed for interactions with a highly avid β-lactamase-tagged basigin prey protein using the AVEXIS assay. The full length (FL) and the major processed form (RH5Ct) of RH5 bound basigin but the N-terminal region (RH5Nt) did not. Bars represent means ± 95% confidence intervals; $n = 3$. (**c**) Biophysical analysis of the RH5Ct-basigin interaction. Serial dilutions of the indicated concentrations of purified RH5Ct protein were injected over the biotinylated ectodomain of basigin immobilized on a streptavidin-coated sensor chip. The data showed excellent fits to a simple (1:1) binding model (red line). The binding parameters of RH5Ct did not differ significantly from those obtained from RH5FL (Supplementary Table 1) demonstrating that the N-terminal region of RH5 does not contribute to basigin binding affinity.

of full-length recombinant *P. falciparum* merozoite 'bait' proteins[24,25]. In total, we screened 39 different parasite proteins which included the known parasite RH5-interacting partners RIPR and CyRPA. We reproducibly identified an interaction with a merozoite protein called P113; no interactions with either RIPR or CyRPA were detected (Fig. 2a). P113 was initially discovered in a study designed to identify GPI-anchored merozoite proteins by purifying detergent-resistant membrane proteins from mature blood-stage parasites[26], and was subsequently found to be one of the most abundant proteins isolated from high molecular mass complexes on the merozoite[27]. P113 has been detected at other stages of the lifecycle including the sporozoite[28] and gametocyte[29], and the inability to select *P113*-deficient parasites suggests that it is essential for blood-stage growth[2]. An informative and rapid validation test for interactions detected using the AVEXIS approach is to determine whether the interaction is dependent on the bait-prey orientation[23]. To establish this, we expressed the entire ectodomain of P113 as an avid prey, and showed that it could interact with both RH5Nt and RH5FL, but not RH5Ct (Fig. 2b). We further demonstrated this interaction by showing that RH5Nt but not control RH5Ct-coated beads could purify native P113 from parasite culture lysates (Fig. 2c). Using antibodies against P113, we could show that it is expressed in early and late-stage schizonts and on the surface of free merozoites by co-staining with the established merozoite surface markers MSP9 (Fig. 2d) and MSP1 (Supplementary Fig.1a), findings that are consistent with an independent study[30]. Antibodies to P113, unlike those to a marker of the inner membrane complex (MTIP), stained both permeabilized and unpermeabilized merozoites demonstrating P113 is surface localized (Supplementary Fig.1b). These data identify P113 as an RH5 interacting partner at the merozoite surface and a new member of the RH5 complex.

**Mapping the P113-RH5 interaction interface**. We next asked which region within RH5Nt contained the P113 binding site. Using a series of truncated RH5Nt recombinant proteins and the AVEXIS assay, we determined a minimal P113 binding region

corresponding to a linear sequence of 19 amino acids that bound P113 indistinguishably from RH5Nt (Fig. 3a). To similarly demonstrate where on P113 the RH5 binding site was located, a series of truncated P113 expression constructs, based on both predicted subtilisin cleavage sites[31] and the location of structurally important cysteine pairings were tested for RH5FL binding (Fig. 3b). The smallest region of P113 that could be expressed, and which retained the ability to bind RH5 encompassed a cluster of 14 cysteine residues at the N terminus of P113 (Fig. 3b). These data identify that the P113 interaction site on RH5 is encompassed within a short linear sequence of just 19 amino acids.

**P113 binds RH5Nt and with higher affinity to RH5FL**. To confirm that RH5 and P113 interact directly and to quantify the biophysical parameters of the interaction, we used SPR. RH5Nt was expressed as a soluble recombinant protein, purified, and serial dilutions injected over P113 immobilized on a sensor chip. Saturable binding was observed demonstrating the specificity of the interaction from which an equilibrium binding constant ($K_D$) of $3.0 \pm 0.5 \mu M$ (mean ± s.d., $n = 2$) was derived (Fig. 4a). Independent kinetic parameters were in agreement with the equilibrium data (Fig. 4b), and although weak, they were within the expected range of many extracellular interactions measured using this technique[32]. Interestingly, the affinity of the interaction was 10-fold higher ($K_D \sim 0.3 \mu M$) when full-length RH5 protein was used as the analyte by comparison to the N-terminal region alone (Fig. 4c). To investigate this in more detail, and remove possible measurement inaccuracies due to the use of different analytes, we measured the biophysical binding parameters in the reverse orientation by using purified P113 as the analyte. We observed that the purified P113 entire ectodomain resolved with a broad profile by size-exclusion chromatography (SEC) with the peak eluting earlier than expected for a monomeric protein suggesting that the ectodomain of P113 self-associated (Fig. 4d). Resolving the fractions corresponding to the elution peak by denaturing and non-denaturing PAGE revealed apparent masses of $\sim 130$ and $\sim 500$ kDa respectively, consistent with the formation of

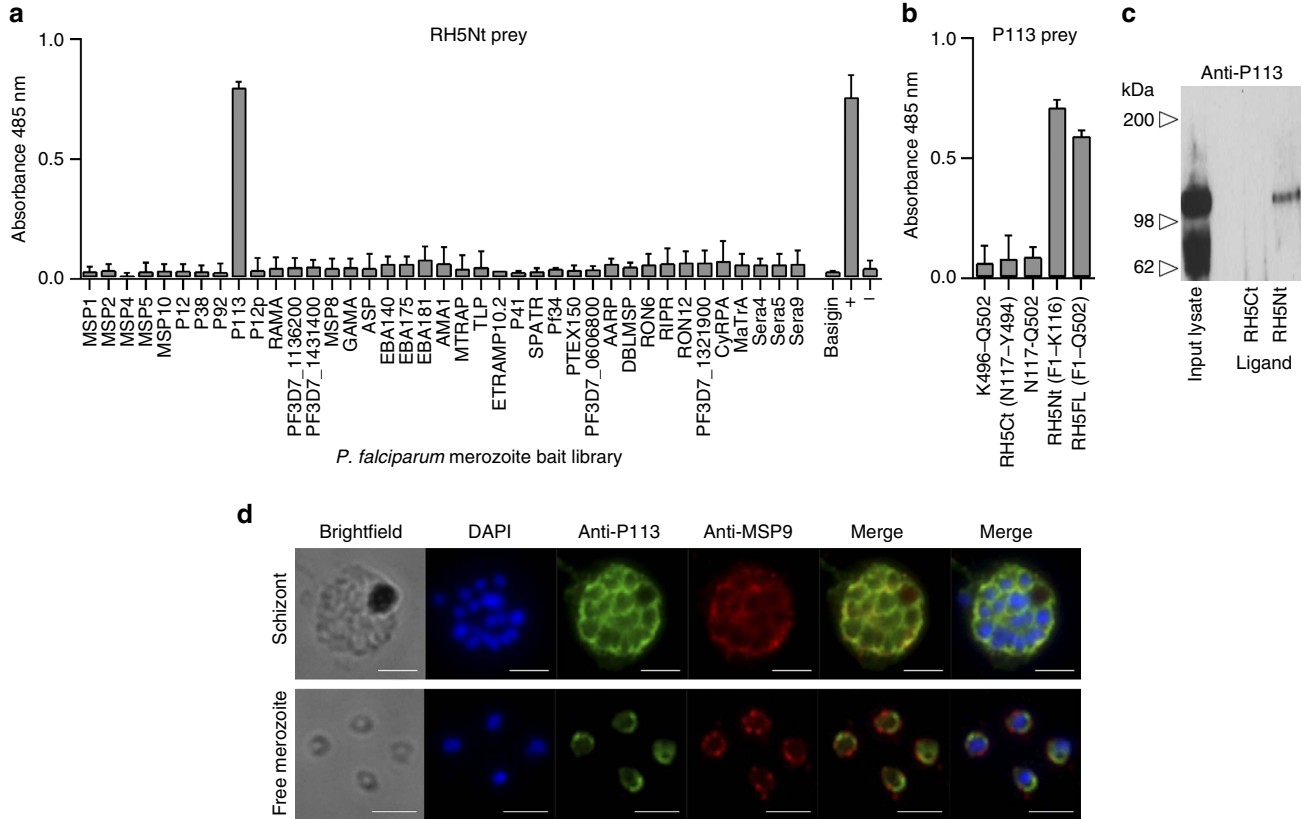

**Figure 2 | The N-terminal domain of RH5 interacts with the merozoite surface protein P113.** (**a**) The N-terminal domain of RH5 (RH5Nt, F1-K116) was expressed as a pentameric, β-lactamase-tagged prey protein and systematically tested for interactions against a library of recombinant *P. falciparum* merozoite receptor ectodomains and secreted bait proteins using the AVEXIS assay; a single interaction with P113 was observed. (**b**) The P113-RH5Nt interaction detected in the reciprocal orientation using a P113 prey against RH5 baits. Bars in (**a**,**b**) represent means ± 95% confidence intervals; $n = 3$; controls were RH5FL-basigin ( + ve) and Cd4d3 + 4 tag alone ( − ve). (**c**) Biochemical purification of native P113 from *P. falciparum* cultures with RH5Nt but not control RH5Ct-coated agarose beads. Parasite lysates were incubated with beads coated in either RH5Nt or RH5Ct protein, washed, and eluates resolved by SDS–PAGE under reducing conditions, blotted and probed with anti-P113 antibody. (**d**) P113 is expressed in schizonts and on the surface of free merozoites. Fixed blood-stage *P. falciparum* schizonts and free merozoites were probed with anti-P113 antibodies (green), the merozoite surface marker anti-MSP9 (red) and nucleic acid stained with DAPI (blue). Scale bars, 3 μm.

tetrameric complexes (Fig. 4e). Analysis of the P113 protein sequence identified a probable coiled-coil region at the C terminus of the protein[33] and similar sequence features have been implicated in the tetramerization of other merozoite proteins such as MSP3[34]. As expected, when the oligomeric P113 complexes were used as analytes in SPR experiments, the shape of the binding curves indicated complex multivalent binding to RH5 immobilized on the sensor chip, which dissociated slowly (Fig. 4f). In an attempt to measure monomeric binding of P113 to RH5Nt for direct comparisons of biophysical binding parameters, we observed that one of the highly-expressed P113 fragments that lacked the predicted coiled-coil region at the C terminus (Y1-N653, Fig. 3b) eluted in much later fractions by SEC (Fig. 4d,e), and when used as the analyte, this truncated form of P113 bound to RH5FL with monophasic binding characteristics and the same ten-fold higher interaction affinity compared to RH5Nt (Fig. 4g, Supplementary Table 1). A kinetic analysis of the interaction confirmed that the higher binding affinity of RH5FL compared to RH5Nt is mainly due to a slower (approximately fourfold) dissociation rate constant, but a higher (approximately twofold) association rate constant also contributes (Supplementary Table 1). We also performed a thermodynamics analysis of the RH5-P113 interaction using SPR which showed that the interaction was entropically driven and that the entropic contribution was higher for RH5FL compared to

RH5Nt providing an explanation for the 10-fold higher affinity of RH5FL for P113 in comparison to RH5Nt (Supplementary Fig. 2). This suggests that the N terminus of RH5 may be partially ordered or in some way structurally constrained when in proximity to the C terminus of RH5, and that the interaction with P113 would release RH5Nt from this restraint thereby causing an overall increase in disorder and consequently a more favourable entropic contribution upon P113 binding. To put these values into context, RH5FL binds P113 with an approximately three-fold higher binding affinity than it binds to the basigin receptor[9], and approximately three-fold lower once processed. Together, these data suggest P113 clusters, probably as tetramers, at the merozoite surface via a membrane-proximal coiled coil region, and binds the N terminus of RH5 with an affinity that is reduced around 10-fold after processing.

**RH5 directly interacts with P113 and CyRPA but not RIPR.** The identification of P113 adds another component to the RH5 invasion complex that is known to include the merozoite proteins CyRPA[21] and RIPR[16], which both co-immunoprecipitate with RH5 from parasite supernatants. Identifying protein complexes using this method does not immediately establish which proteins directly interact with each other, nor whether the components can bind simultaneously. To address these questions, we used our

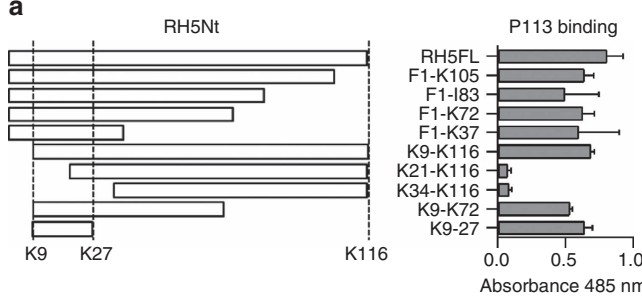

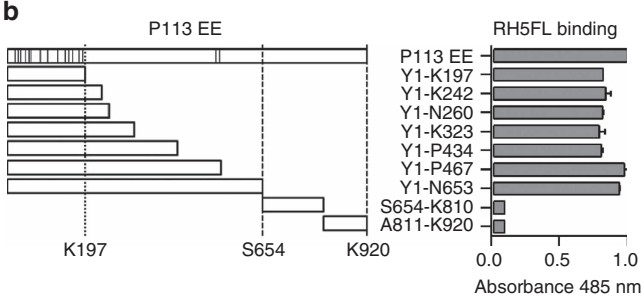

**Figure 3 | The P113-RH5 interaction interface involves a short linear sequence in RH5Nt and the cysteine-rich N-terminal domain of P113.**
(**a**) The P113 interaction interface on RH5 was mapped to 19 amino acids located within the N terminus of RH5. Fragments of RH5Nt that could be expressed as prey proteins are depicted schematically and were tested for their ability to bind a P113 bait protein by AVEXIS. The K9-K27 fragment represented the minimal binding region. (**b**) The RH5 interaction interface on P113 was mapped to the N terminus corresponding to a cysteine-rich region encompassed by Y1-K197. Fragments of P113, including the entire ectodomain (EE), that could be expressed as bait proteins are depicted schematically as in **a** with the approximate locations of the 16 cysteines marked by vertical bars. The proteins were tested for their ability to bind an RH5 prey protein by AVEXIS. Bars represent means ± 95% confidence intervals $n = 3$; the experiment shown is one of two independent experiments.

collection of recombinant *P. falciparum* merozoite proteins expressed in mammalian cells[24,25] together with the AVEXIS assay. Previously, we have shown that direct binary interactions can be detected using this approach between RH5FL/Ct and basigin (Fig. 1b and[9]) as well as RH5FL/Nt and P113 (Fig. 2b). Screening the merozoite protein library with an RH5FL pentameric 'prey' identified the expected interactions with both P113 and CyRPA baits demonstrating direct binding of RH5 with each of these proteins (Supplementary Fig. 3a). We additionally determined that the CyRPA binding site on RH5 was located in the RH5Ct fragment since RH5Ct but not RH5Nt bound CyRPA (Supplementary Fig. 3b). The same interactions were detected when the bait-prey orientations were reversed with RH5FL presented as a monomeric 'bait'—an important validation criteria used for AVEXIS assays (Supplementary Fig. 3c). We did not observe any other interactions with RH5FL, including RIPR (Supplementary Fig. 3a,c). CyRPA, however, did directly interact with RIPR (Supplementary Fig. 3b) suggesting that CyRPA bridges the interaction between RH5 and RIPR. No direct interactions were detected between P113 and either CyRPA or RIPR (Supplementary Fig. 3d). Altogether, these data establish that P113 and CyRPA bind directly to RH5 while RIPR is associated with RH5 through a mutual interaction with CyRPA.

**RH5-CyRPA-RIPR complex interacts with basigin but not P113.**
Given the essential role of the RH5 complex in invasion, we sought to address the interaction dependencies of each component of the RH5 complex with one another. We again took advantage of our recombinant proteins and AVEXIS assay, but this time attempted to induce binding between non-interacting baits and preys by titrating in additional components of the complex. We first established that a basigin prey could be captured by a P113 bait by adding increasing concentrations of purified monomeric RH5FL, but not RH5Ct or RH5Nt (Fig. 5a). This demonstrated that RH5FL was able to bind both P113 and basigin simultaneously, consistent with them interacting with distinct regions of RH5. Similarly, RH5FL (but not RH5Ct or RH5Nt) could form a trimeric complex involving CyRPA and P113 (Fig. 5b). RH5FL and RH5Ct—but not RH5Nt—could interact simultaneously with CyRPA and basigin indicating that their respective interaction interfaces do not overlap in the C-terminal region of RH5 (Fig. 5c). Consistent with all these data, the formation of the basigin-RH5-P113 complex could be detected in the presence of CyRPA (Fig. 5d). We next showed that an RH5FL prey could be captured by a RIPR bait only by adding increasing concentrations of purified CyRPA (Fig. 5e), which provided an explanation for why RIPR did not directly interact with RH5 but is co-purified with RH5 from parasite supernatants[16]. We next asked whether the complex of the three parasite proteins RH5, CyRPA and RIPR could interact with either the erythrocyte or merozoite-tethered receptors. Interestingly, while an interaction could be detected with basigin, we were unable to detect interactions with P113 (Fig. 5f). Altogether, these results suggest that RH5-CyRPA-RIPR complexes containing either RH5FL or RH5Ct can bind the erythrocyte receptor basigin, but not the merozoite tethering molecule P113.

During the preparation of this manuscript, a report describing the RH5-CyRPA interaction was published showing that immunoprecipitated CyRPA was labelled by the addition of radioactively labelled glucosamine and mannose, suggesting that CyRPA is post-translationally modified by addition of a GPI anchor[21]. To try and confirm this, we took advantage of the fact that *P. falciparum* proteins can be appropriately modified with GPI anchors when expressed in mammalian cells[35]. By contrast to P113—which was used as a positive control—we did not find any evidence that CyRPA was GPI-anchored and surface localized (Supplementary Fig. 4). Our findings are consistent with the initial prediction that CyRPA is a secreted protein[22] and the absence of CyRPA-derived peptides from detergent-resistant *P. falciparum* schizont membrane fractions that are enriched in GPI-anchored proteins[27]. During the late stages of revising this manuscript, another study has directly shown that CyRPA is not GPI-anchored[36].

**An RH5Nt amph-vaccine elicits invasion-blocking antibodies.**
Because the RH5 N-terminal region is relatively short and predicted to be unstructured, we asked whether a chemically synthesized peptide could elicit invasion-blocking antibodies and therefore be a component of a cost-effective vaccine. A peptide corresponding to the first 116 amino acids of RH5 was synthesized with an additional C-terminal cysteine for chemical coupling. We first showed that the synthetic peptide structurally mimicked RH5Nt by showing it retained the ability to bind P113 with almost identical biophysical binding parameters to RH5Nt (Fig. 6a; Supplementary Table 1). Conjugating peptides to amphiphilic compounds that contain both hydrophobic albumin-binding acyl lipid chains and a solubilizing hydrophilic PEG spacer have been shown to improve immune presentation by promoting lymph node accumulation through 'albumin hitchhiking'[37]. We therefore covalently coupled the peptide corresponding to RH5Nt to a maleimide-functionalized

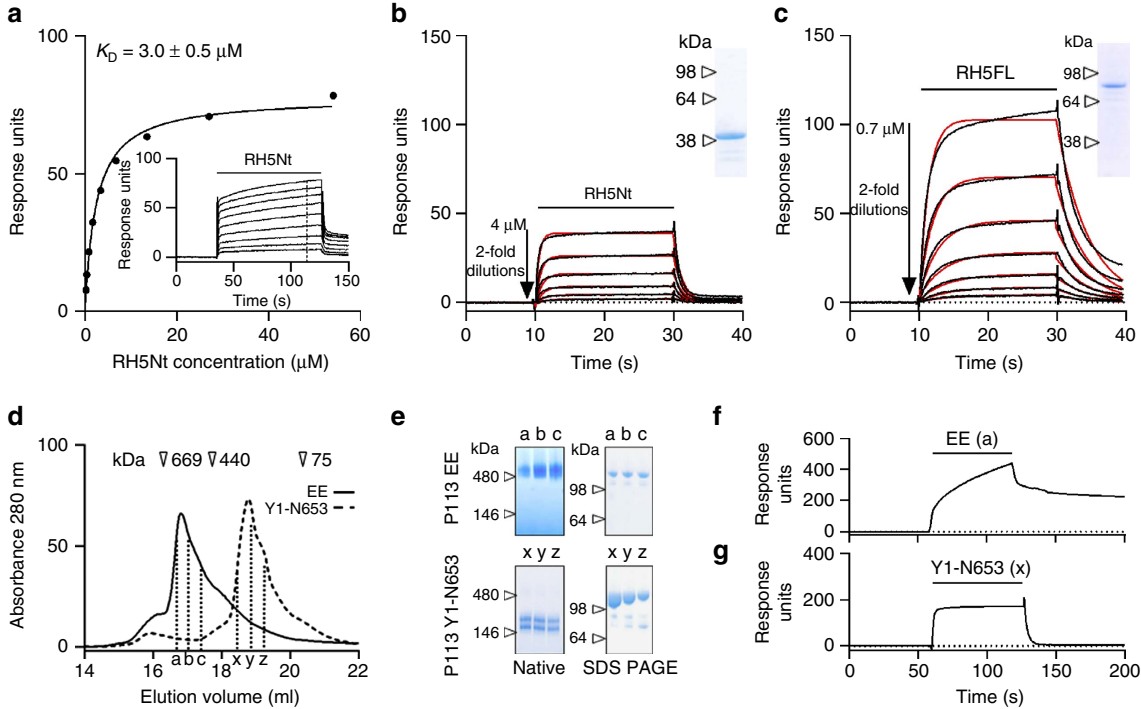

**Figure 4 | The interaction between P113 and the N-terminal region of RH5 is specific and 10-fold stronger in the context of full-length RH5.** (**a**) RH5Nt bound P113 specifically and with low affinity. Purified monomeric RH5Nt was serially diluted before being injected over 780 RU of monobiotinylated P113 immobilized on a streptavidin-coated sensor chip. The binding traces suggested the presence of a small amount of multimeric material (see Methods); consequently, binding equilibrium was not quite achieved even though long ( > 60 s) injection times were used; the dashed line (inset) marks the time point used for equilibrium binding. From these data, an equilibrium binding constant ($K_D$) of $3.0 \pm 0.5\,\mu M$ (mean $\pm$ s.d., $n = 2$) was calculated. (**b,c**) RH5FL binds P113 with a ten-fold higher affinity than RH5Nt. A kinetic analysis of P113 binding RH5Nt (**b**) or RH5FL (**c**) using SPR; inset gels demonstrate the homogeneity and integrity of each RH5 analyte preparation. Serially diluted RH5 analytes were injected over 780 RU of immobilized P113. Binding data (black lines) fitted a simple 1:1 binding model well (red lines), and were used to determine binding constants. (**d**) Gel filtration elution profiles of the entire ectodomain (EE) and an N-terminal subfragment (Y1-N653) of recombinant tagged P113 demonstrated that P113 EE can form multimers. The lettered dashed lines correspond to fractions collected over the resolved peaks; the elution volumes of gel filtration mass markers are indicated. (**e**) Native and denaturing (SDS–PAGE) gels corresponding to aliquots of the fractions indicated by the letters in **d**. P113 EE but not the Y1-N653 fragment resolves at an approximately fourfold higher mass under native when compared with denaturing conditions. The Y1-N653 fragment resolved as two species by native PAGE. Multimeric forms of P113 EE exhibited complex multivalent binding behaviour by SPR (**f**) compared with P113 Y1-N653 (**g**). Sensorgrams in (**f,g**) show a 120 s pulse of purified $8\,\mu M$ P113 analyte injected over 510 RU of immobilized RH5Nt.

PEG$_{2000}$-DSPE molecule (Fig. 6b) to create an amph-vaccine[37]. Polyclonal antibodies against this compound elicited good antibody titres (Supplementary Fig. 5) that prevented RH5 binding to P113 (Fig. 6c), but not basigin (Fig. 6d). The antibodies inhibited parasite growth *in vitro*, an effect that was observed in both 3D7 and Dd2 parasite strains, and specifically affected schizont to ring stage progression demonstrating that they inhibited erythrocyte invasion (Fig. 6e and Supplementary Fig. 6). These data suggest that a peptide based on the N-terminal region of RH5 would be an effective and inexpensive component of a malaria vaccine.

## Discussion

RH5 is a strong candidate for inclusion in a malaria vaccine because it is susceptible to vaccine-elicited antibodies and its interaction with the basigin receptor is essential for invasion. The finding that RH5 interacts with both host and parasite proteins has led to a model whereby RH5 functions as a central organizing component of a protein complex formed at the nexus of the merozoite–erythrocyte interface that triggers further rhoptry release and the formation of an 'open connection' through which other invasion proteins can be delivered to establish the moving junction[4]. One missing component of this model is a molecular

explanation for how the secreted RH5 protein is localized to the merozoite surface. Here we use a library of recombinant proteins and a sensitive assay designed to detect low-affinity extracellular interactions to identify the GPI-anchored merozoite surface protein P113 as a direct interacting partner of RH5 which could provide such a tethering mechanism. Importantly, we show that the P113 interaction site on RH5 lies within the N terminus of RH5—a region which is absent from the major form of RH5 that is detected in parasite culture supernatants, and therefore provides a regulatory mechanism by which RH5 could be released from the merozoite surface during invasion. Moreover, by using individual components of the RH5 complex, we have determined that not all constituents of the RH5 complex can interact simultaneously, again suggesting that there is an ordered schedule of protein interactions that could delineate separable molecular events during invasion. Adding the contributions described here into a broader context compiled from many other researchers suggests the following model (Fig. 7).

Proteomics studies of *P. falciparum* blood-stage membrane preparations have shown that P113 is GPI-anchored and one of the most abundant proteins constitutively present on the merozoite surface[26,27,38], which we show here is likely to be clustered on the surface as tetramers. There are few studies on *P. falciparum* P113, but antibody responses against P113 have

been shown to be associated with protection from malaria in study cohorts from both Africa[39] and Papua New Guinea[40]. Although it is unclear precisely when RH5 is proteolytically cleaved during the rapid invasion process, it is tempting to speculate that at least a proportion of RH5 is released from rhoptries as an unprocessed full length form and confined to the merozoite surface via direct interactions between its N-terminal region and P113 with a relatively high ($K_D = 0.3\,\mu M$) binding affinity; the clustering of P113 may be important for increasing local binding avidity for RH5FL. We show here that RH5FL can simultaneously bind P113 and basigin through the N- and C-terminal regions, respectively, so it is molecularly possible for RH5 to bridge the merozoite and erythrocyte membranes through these interactions. The P113-RH5-basigin complex may then trigger secretion of other parasite ligands including the AMA1/RON complex which lead to the formation of the moving junction[4]. We have established here that the secreted CyRPA protein can bind RH5 within the P113-RH5-basigin complex, and because we could detect direct interactions between RIPR and CyRPA but not RH5, CyRPA may perform the role of recruiting RIPR to the RH5 complex. These findings provide two possible mechanisms for RH5 complex release from P113 at the merozoite surface: (1) the proteolytic cleavage of the N-terminal region of RH5; and (2) the mutually exclusive binding of RIPR and P113 to the RH5 complex, which suggests that the recruitment of RIPR to the complex would allow its release from the merozoite surface. In summary, we suggest that at the moment of invasion, RH5 is locally concentrated and presented to the basigin receptor at the site of contact between the merozoite and erythrocyte through interactions with surface localized P113. The binding of the CyRPA and RIPR proteins to RH5 may act as a one-way 'switch' to licence the next step of the invasion process by releasing the RH5 complex from the surface-tethered P113 due to the mutually exclusive binding of RIPR and P113 in the RH5 complex, and/or recruitment of a protease that cleaves the N-terminal region of RH5. The identification of the protease/s responsible for RH5 cleavage will be an important step in understanding the function of the RH5 invasion complex. Whilst the observation that aprotinin can reduce RH5 processing suggests the involvement of a serine protease, RH5 is not a predicted or experimentally validated substrate of the well-characterized parasite blood-stage serine proteases, the subtilisins including SUB1[31].

One intriguing evolutionary puzzle is that clearly identifiable orthologues of each component of the RH5 complex do not necessarily co-occur in the genomes of different *Plasmodium* species: orthologues of both *P113* and *RIPR* are found across *Plasmodium* species that infect both primates and rodents; *CyRPA* orthologues are present in primate, but not the rodent parasites; and *RH5* is further restricted to a subset of the primate parasites closely related to *P. falciparum* within the subgenus

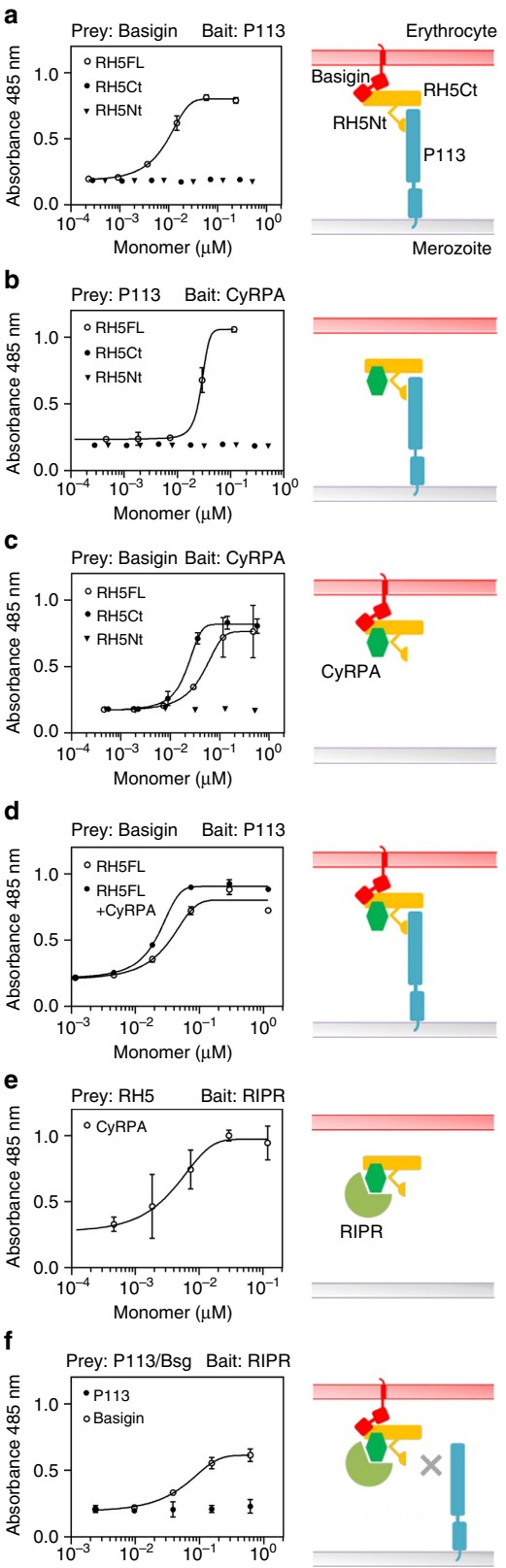

**Figure 5 | The RH5-CyRPA-RIPR complex can interact with basigin but not P113.** The binding interdependencies of the proteins within the RH5 invasion complex with basigin and P113 were determined using modified AVEXIS assays. The indicated purified monomeric components of the RH5 complex were titrated into binding reactions between the named baits and β-lactamase-tagged preys, which do not interact directly, and any resulting prey binding quantified by measuring the hydrolysis of a colorimetric β-lactamase substrate at 485 nm. Binding data are shown on the left panels with their interpretations shown schematically on the right. RH5FL can simultaneously bind P113 and basigin (**a**), CyRPA and P113 (**b**), and CyRPA and basigin (**c**). (**d**) The basigin-RH5FL-P113 complex is not overtly affected by the addition of CyRPA; here, the RH5FL monomer concentration was titrated with CyRPA held constant at 0.3 μM. (**e**) RH5FL prey could be captured on a RIPR bait by addition of purified CyRPA. (**f**) The RH5FL-CyRPA-RIPR complex interacted with basigin, but not P113 preys; here, the CyRPA monomer concentration was titrated with RH5FL held constant at 0.2 μM. Binding data points represent means ± 95% confidence interval ($n = 3$); a representative experiment from at least two independent experiments is shown.

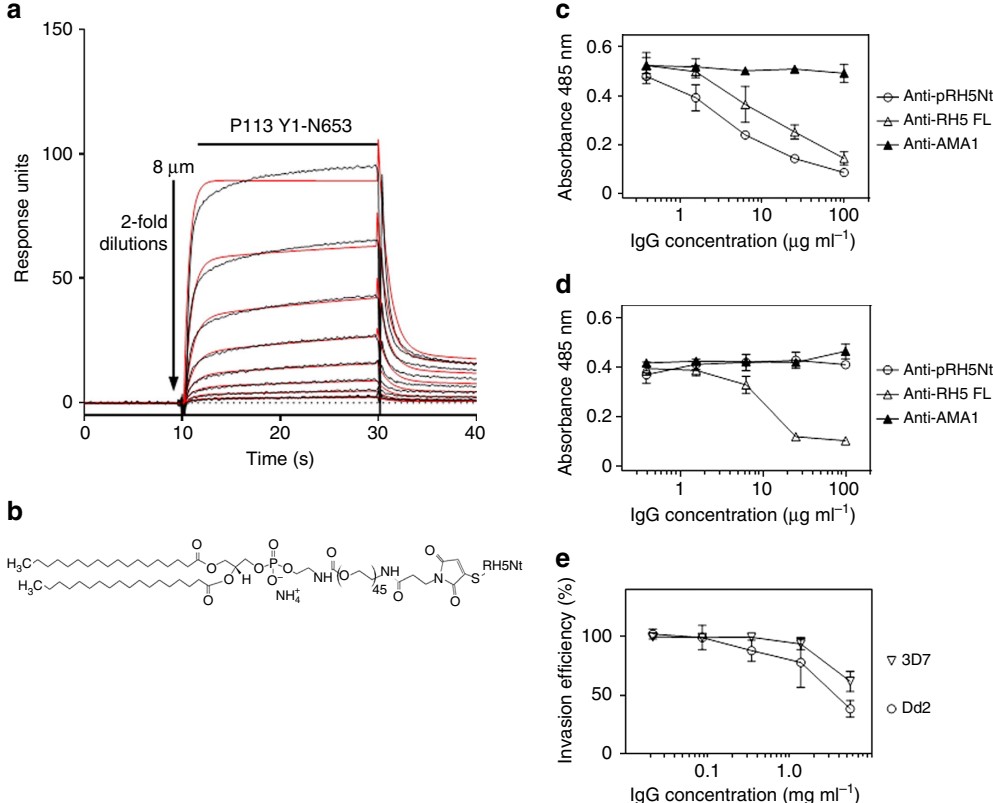

**Figure 6 | An 'amph-vaccine' based on RH5Nt elicits antibodies that inhibit parasite growth *in vitro*. (a)** A synthetic 116 amino-acid peptide corresponding to RH5Nt interacts with P113. Serial dilutions of purified P113 (Y1-N653) were injected over the RH5Nt peptide immobilized on a streptavidin-coated sensor chip after the C-terminal cysteine was conjugated to biotin functionalized with maleimide. Reference-subtracted binding data are shown (black lines) which fit well to a 1:1 binding model (red lines). **(b)** Structure of an 'amph-vaccine' based on RH5Nt created by conjugating the RH5Nt peptide to maleimide-functionalized 1,2-distearoyl-sn-glycero-3-phosphoethanolamine-N-[maleimide(polyethylene glycol)-2000] (PEG$_{2000}$-DSPE). Antibodies raised against RH5Nt peptide (Anti-pRH5Nt) blocked RH5 binding to P113 **(c)**, but not to basigin **(d)**. The indicated concentrations of protein-G purified rabbit polyclonal antibodies were incubated with RH5 β-lactamase-tagged prey proteins before presenting them to immobilized P113 **(c)** or basigin **(d)** baits. Prey binding was quantified by nitrocefin hydrolysis at 485 nm; polyclonal antibodies to RH5FL and AMA1 were used as positive and negative controls respectively. **(e)** Polyclonal antibodies elicited against the RH5Nt amph-vaccine inhibited erythrocyte invasion of both 3D7 and Dd2 strain of *P. falciparum*. Data points represent means ± 95% confidence interval, $n = 3$; a representative experiment from two independent experiments is shown.

*Laverania*. Most likely, the proteins encoded by these genes perform different roles in these species but have been co-opted by *P. falciparum* together with the evolution of the RH5 protein for their roles in invasion. Functional studies in *P. berghei*, for example, have shown that *P113*-deficient parasites could complete their lifecycle, but had lower numbers of parasites making the transition from sporozoites to the liver-stage form[41]; and, more recently, P113 has been shown to be a peripheral member of the PTEX translocation machinery which is conserved across *Plasmodium* spp.[42].

Finally, we show that a linear peptide corresponding to the N terminus of RH5 retains the functional ability to bind P113 and, importantly, could elicit antibodies that inhibit parasite growth. This is of interest because peptides can be synthesized at scale and are chemically defined—both relevant considerations for the production of a vaccine. Recent and continuing developments in adjuvant-enhancement of peptides to safely elicit high-titre antibody responses such as the use of albumin-binding amphiphilic molecular groups may present new opportunities in the development of cost-effective vaccines. An affordable and highly effective malaria vaccine would undoubtedly be an important control measure to improve the lives of the billions of people who live at risk of this deadly infectious disease.

## Methods

**Protein-based methods.** Recombinant *P. falciparum* proteins were expressed by transient transfection of HEK293 cells from the 3D7 strain using expression plasmids described in refs 24,25. With the exception of RIPR that was expressed in Freestyle 293-F cells (ThermoFisher Scientific), all proteins were expressed using HEK293-6E cells[43]. RH5 was processed (Fig. 1a) when expressed in HEK293 media supplemented with fetal calf serum; this processing was reduced by using HEK293 cells adapted to serum-free media and prevented where necessary by the addition of (2 to 10 µg ml$^{-1}$) aprotinin. Briefly, all plasmids were chemically synthesized using codons optimized for expression in human cells, potential N-linked glycosylation sequons were mutated, and contained a C-terminal rat Cd4d3 + 4 tag. Where appropriate, monomeric 'bait' proteins were enzymatically monobiotinylated by cotransfection with a plasmid encoding a secreted BirA enzyme as described[23,44]. His-tagged proteins were purified from supernatants on HisTrap HP columns using an ÄKTAexpress or ÄKTApure (GE Healthcare) and resolved by gel filtration on a Superdex 200 Increase 10/300 column (GE Healthcare) as described[45]. Regions encoding defined domains of RH5 or P113 were amplified by PCR from plasmids encoding the entire ectodomain and cloned into appropriate expression plasmids. Boundaries for each of the recombinant protein fragments are indicated using amino-acid numbers pertaining to the *P. falciparum* 3D7 sequence without signal peptide. Recombinant proteins were resolved on Bis-Tris 4–12% SDS polyacrylamide gels and stained with colloidal Coomassie blue. Where necessary, proteins were blotted on to Amersham HyBond PVDF membranes by transfer in NuPAGE transfer buffer (ThermoFisher Scientific) for 1 h at 40 V; membranes were blocked in PBS/2% BSA for 3 h, probed with 1:10,000 primary antibodies, washed four times with PBST (PBS/0.1% Tween-20), and then incubated with 1:5,000 of anti-rabbit horseradish peroxidase-conjugated secondary (ThermoFisher Scientific Cat. No. G21234). After further washing four times in PBST blots were exposed on film using West Pico ECL

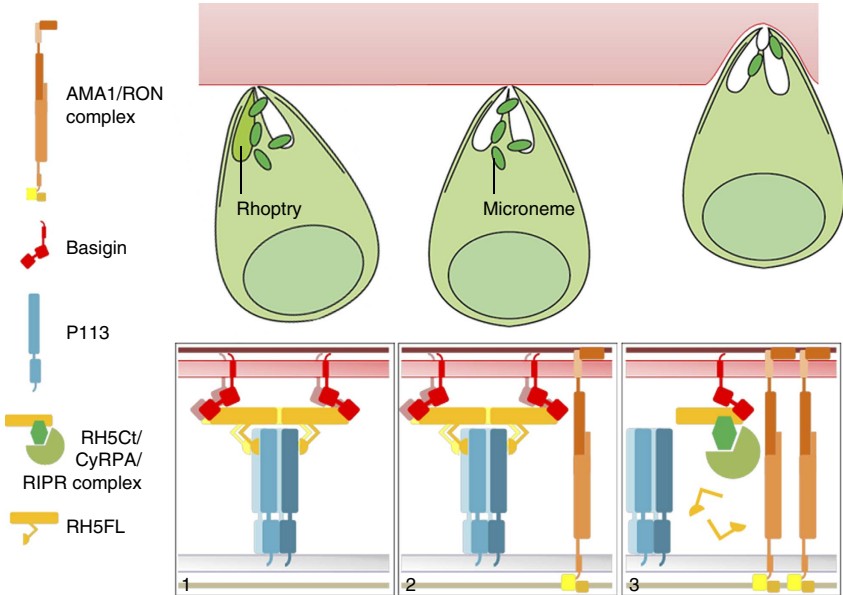

**Figure 7 | A model describing the role of the RH5 complex and its interaction with P113 during erythrocyte invasion.** Before invasion, the different components of the RH5 invasion complex are segregated within different subcellular locations of the merozoite and therefore purposefully prevented from interacting: RH5 is located in the rhoptries, CyRPA and RIPR in the micronemes, and P113 on the surface of the merozoite. Following engagement of the erythrocyte and release of the rhoptry contents, RH5 is tethered via its N-terminal region at the merozoite membrane by the surface-localized multimeric P113, enabling direct presentation to the basigin receptor on the erythrocyte surface and leading to the formation of an open connection for other invasion ligands to be secreted (1). The AMA1/RON complex initiates the formation of the moving junction (2). The localized secretion of CyRPA and RIPR from the micronemes leads to the formation of the RH5-CyRPA-RIPR-basigin invasion complex which, either because of the cleavage of RH5 to remove the N-terminal region, and/or because P113 and RIPR cannot simultaneously bind the RH5 complex would release RH5 from the P113 tether at the merozoite surface to licence invasion (3). The P113-RH5 complex would therefore only be fleetingly formed during the rapid invasion process resulting in a soluble post-invasion RH5Ct-CyRPA-RIPR complex.

(ThermoFisher Scientific). Uncropped gels and blots used to produce Figs 1a, 2c and 4e and Supplementary Fig. 4 are provided in Supplementary Fig. 7. Recombinant proteins analysed by SEC were resolved by native PAGE using Bis-Tris polyacrylamide gels at 150 V and colloidal Coomassie staining. Edman N-terminal protein sequencing was performed by the PNAC facility at the University of Cambridge using standard procedures.

**Avidity-based extracellular protein interaction screening.** AVEXIS was performed essentially as described[23]. Monomeric biotinylated bait proteins and highly avid pentameric β-lactamase-tagged prey protein activities were prepared and their expression levels normalized as described[44] to ∼5 µg ml$^{-1}$ before their use in interaction screening. Briefly, biotinylated baits were immobilized in streptavidin-coated 96-well microtitre plates, washed with PBST (PBS/0.1% Tween-20), incubated with prey proteins, washed three times with PBST and any captured preys quantified by adding the colorimetric β-lactamase substrate nitrocefin and measuring the absorbance of the hydrolysis products at 485 nm. The negative control in each screen was the query prey protein probed against the Cd4d3 + 4 tag alone. Positive control interactions included the human basigin bait detected with the RH5 prey, and the rat Cd200 bait with rat Cd200R prey.

**SPR.** SPR was performed using a Biacore T100 instrument (GE Healthcare) as previously described[45]. Briefly, ∼150 RU of biotinylated rat Cd4d3 + 4 was captured on a streptavidin-coated sensor chip (GE Healthcare) to be used as a reference, and an equivalent molar quantity of query protein was captured in the query flow cell. Immediately before SPR experiments, analyte proteins were separated by size exclusion chromatography to remove aggregates, which even in very small amounts can affect kinetic measurements[46]. Although monomeric RH5Nt resolved as a monodisperse peak by gel filtration at the expected size, the initial binding kinetics of the RH5Nt-P113 interaction were initially rapid followed by much slower binding kinetics, and this behaviour was mirrored in the dissociation phase once the injection of RH5Nt was complete, suggesting the presence of small amounts of active multimeric material; attempts were made to reduce this affect by performing SPR immediately after size separation. Kinetic studies were performed at high flow rates (100 µl min$^{-1}$) to reduce confounding rebinding effects and equilibrium binding analysis at 10 µl min$^{-1}$. Where necessary, surfaces were regenerated by injecting a pulse of 2 M NaCl. The retention of immobilized ligand surface activity was confirmed by showing equivalent binding responses of the same concentration before and after chip

regeneration. Binding data were analysed in the Biacore T100 evaluation software (GE Healthcare) using reference-subtracted sensorgrams. Independent protein preparations of both analyte and ligand proteins were used for replicate experiments, all of which were performed at 37 °C.

**RH5Nt amph-vaccine and antibodies.** A 117 amino-acid peptide corresponding to RH5Nt (F1-K116) with an additional C-terminal cysteine was chemically synthesized (LifeTein). The peptide was dissolved in DMF and conjugated to maleimide-PEG2000-DSPE (Avanti Polar Lipids) at a 1:2 molar ratio overnight at room temperature with rotation. Polyclonal antisera were raised by immunizing rabbits six times with 100 µg of purified antigen in Freund's adjuvant in rabbits (Cambridge Research Biochemicals) in strict accordance with UK Home Office governmental regulations and the local Sanger Institute animal welfare ethical review board. Polyclonal antisera were similarly raised against full-length CyRPA and the P113 Y1-N653 fragment expressed in mammalian cells as rat Cd4d3 + 4-6His tagged proteins and purified using Ni$^{2+}$-NTA chromatography. The sera were purified according to manufacturer's instructions on an ÄKTA system using separate Hi-Trap Protein G HP columns (GE Healthcare) for each rabbit serum. The fractions were eluted with 0.2 M glycine (pH 2.7) directly into a neutralization buffer (1 M Tris:Cl pH 9.0) and immediately dialysed into PBS.

***P. falciparum* culture and invasion assays.** The 3D7 and Dd2 *P. falciparum* strains were obtained from BEI Resources and cultured in human O + erythrocytes at 5% haematocrit in complete medium (RPMI-1640 containing 10% Albumax under an atmosphere of 1% $O_2$, 3% $CO_2$ and 96% $N_2$. Blood was purchased from UK NHS Blood and Transplant (NHSBT) services. Consent from healthy volunteers was obtained by NHSBT using a study protocol approved by the National Research Ethics Service Committee East of England—Cambridge South and by the Sanger Institute Human Materials and Data Management Committee; samples were provided fully anonymized. Invasion assays were carried out in round-bottom 96 well plates, with a culture volume of 100 µl per well at a haematocrit of 2%. Parasites were synchronized at early stages with 5% (w/v) D-sorbitol; trophozoite stage parasites were mixed with blocking agent and then incubated in the plates for 24 h at 37 °C inside a static incubator culture chamber. At the end of the incubation period erythrocytes were collected and parasitized erythrocytes were stained with 2 µM Hoechst 33342. Purified antibodies were dialysed into RPMI (GIBCO) before use. Hoechst 33342 stained samples were excited with a 355 nm UV laser (20 mW) on a BD LSRH flow cytometer

(BD Biosciences) and detected with a 450/50 filter. BD FACS Diva was used to collect 100,000 events for each sample. FSC and SSC voltages of 423 and 198, respectively, and a threshold of 2,000 on FSC were applied to gate the erythrocyte population. The data collected were further analysed with FlowJo (Tree Star). All experiments were carried out in triplicate.

**Immunofluorescence of *P. falciparum*.** Air-dried thin films of late-stage schizonts and free merozoites were fixed in ice-cold methanol. Fixed parasites were permeabilized in PBS/Triton X100 1% (except in Supplementary Fig. 1b) and blocked in PBS/3%BSA/10%goat serum. Parasites were stained with the primary antibodies: 1:10,000 rabbit anti-P113 polyclonal (this study), 1:400 mouse anti-MSP9 polyclonal (a gift of Alexander Douglas[11]), 1:5,000 mouse anti-MSP1 monoclonal (a gift of Michael Blackman), 1:1,000 rat anti-MTIP[47] overnight at 4 °C. After three washes, the parasites were incubated with secondary antibodies (goat anti-rabbit IgG (H + L) AlexaFluor 488, anti-mouse IgG (H + L) AlexaFluor 555, goat anti-rat IgG (H + L) AlexaFluor 555, all ThermoFisher Scientific Cat. No.s A-11008, A-21424 and A-21434, respectively) diluted 1:500 for 1 h at room temperature. The samples were washed in PBS three times and mounted in Prolong Gold (Molecular Probes) with DAPI before images were captured on a Leica DMi8 fluorescent microscope and processed using Leica LAS X software and Photoshop.

**P113 purification from parasite cultures.** *P. falciparum* parasites were tightly synchronized by alternating treatment between 5% D-sorbitol and isolation on a Percoll gradient. Erythrocytes containing mature schizonts from 50 ml cultures (2.5% haematocrit and 2 to 7% parasitaemia) were isolated on a Percoll gradient and lysed with 0.15% saponin. The parasite pellet was washed with PBS and resuspended in lysis buffer (25 mM Tris-Cl, pH 7.4, 1 mM EDTA, 1% NP40, 150 mM NaCl, 5% glycerol, 5 mM AEBSF, 4 mM aprotinin, 0.2 mM bestatin, 75 μM E64, 0.1 mM Leupeptin, 50 μM Pepstatin A). The release of P113 into the aqueous phase was improved by either the addition of 1% Triton X-100 or vortexing and rotation at room temperature for 1 to 4 h. The lysate supernatant was incubated with streptavidin-conjugated agarose beads for 1 h at room temperature to remove non-specific binders. Streptavidin agarose beads (Pierce) were loaded with monomeric biotinylated RH5Ct or RH5Nt (as per manufacturer's instructions) and added to the lysate supernatant (∼10^9 parasites per 100 μg of RH5Ct or RH5Nt protein). After incubating for 4 h at room temperature the agarose beads were washed three times (PBS, 0.1% Tween 20) and the bound proteins eluted in 2% LDS. The eluate was analysed by SDS–PAGE under reducing conditions and probed with P113 polyclonal antibodies.

**Cell surface expression of P113.** Plasmids containing the CyRPA and P113 coding sequences were chemically synthesized as described above for the merozoite cell surface library but with three differences: constructs included their endogenous signal peptides, the GPI-sequence of P113 was retained, and the proteins lacked any protein tags; these plasmids were transiently transfected in HEK293 cells. After 24 and 48 h the culture supernatant and cells were collected, probed with the appropriate polyclonal IgG raised against CyRPA or P113 and stained with anti-rabbit IgG (H + L) conjugated to Alexa Fluor 568 (ThermoFisher Scientific Cat. No. A10042). PBS was used to wash the cells between antibody incubations. Flow cytometry was performed on a BD LSR Fortessa.

**Data availability.** All relevant data are available from the authors upon request.

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

## Acknowledgements

This work was supported by the Wellcome Trust grant number 098051 and Medical Research Council (G1000413), a Sanger Institute translational award to F.G. and G.J.W. We would like to thank Cecile Crosnier (Sanger Institute, UK) for the initial Edman sequencing of cleavage sites, Paula Cawkill and Sarah J. E. Marsden (Sanger Institute, UK) for assisting with parasite culture and invasion assays, and Alexander Douglas (Sanger Institute, UK) and Michael Blackman (Crick Institute, UK) for providing the MSP9 and MSP1 antibodies. The following reagent was obtained through BEI Resources, NIAID, NIH: *Plasmodium falciparum*, Strains 3D7, MRA-102, contributed by Daniel J. Carucci and Dd2, MRA-150, contributed by David Walliker.

## Author contributions

F.G. performed all the experiments except identification of the RH5-P113 interaction sites which was performed by M.F. under the guidance of F.G., and parasite immuno-fluorescence which was performed by L.G.D. Parasite invasion assays and culture were performed by N.C. and A.C.K. under the guidance of J.C.R. The study was conceived and the manuscript written by F.G. and G.J.W.

## Additional information

**Competing financial interests:** The authors declare no competing financial interests.

