## [Peer Review File · Nature Communications]

Reviewers' comments:

Reviewer #1 (Remarks to the Author):

The paper by Galaway and Colleagues provides an exciting extension to the pioneering use of the AVEXIS screening technology by the Wright lab to the question of receptor and ligand-complexes involved in the key process of malaria parasite invasion of the human erythrocyte.

In this paper, the authors identify a surface bound, GPI anchored protein, previously called P113 (or PF14_0201/PF3D7_1420700), which in vitro interacts with the lead antimalarial blood-stage vaccine candidate Rh5. The data provided presents good evidence for the binding of P113 to Rh5 and proposes a model, derived from binding studies between Rh5 and its other binding partners (CyRPA and RIPP), for a proteolytically-regulated Rh5 complex binding to either parasite surface, RBC surface or between both. Another key observation by the paper is resolving that CyRPA is not GPI anchored - data that I find quite convincing.

Core Critique

The key gap in the paper is in vivo parasite proof that supports the interaction between P113 and Rh5 (and the complex) and provides a native framework when the interaction takes place (i.e. shows its actually important/relevant/real). With several older published studies on P113, I would have hoped that antibodies could be sourced that would allow both the co-immuno-precipitation of the complex (as was done with RIPP and CyRPA) to verify (or at least attempt to verify) that this interaction is indeed confirmed in vivo and just as importantly (or even more) whether the timing of events fits with the model proposed. A core concern is that without such data, i.e. simply relying on the protein-protein work that this could all be an artefact of the in vitro nature of the work (though I am not necessarily suggesting that it is). Memories of aldolase and TRAP-like protein tails come to mind!

For example, the timing of the interaction (and spatial localisation of proteins involved) would appear to me to be the key to how this would or would not work. E.g. How does the complex with RIPP NOT come together? If RIPP is a micronemal protein it would be secreted at the same time or even prior according to current understanding so it would be around. Thus complex formation must be regulated by the unknown protease - but how? Similarly, when is Rh5 actually processed? Some proteins are pre-processed before they are even released. If it is already in a processed form in the schizont, then it might be that P113 binding is already compromised before invasion even begins - not so good for the model and not so good for a vaccine either.

Specific Questions:

Q1. What is the mechanism/basis for 113 binding to the full length of Rh5 with a 10 fold higher affinity than the processed N terminus? Presumably there must be some structural change or masking in the processed form? Or is the binding site partly covered by both regions N and C terminal?

Q2. The activity of antibodies elicited by the synthetic peptide of Rh5 is interesting. However, the activity appears to be quite weak (in the 1-2+ (may be even 10) mg range). This would not appear to me to be potent enough to be a serious contender for an inhibitory antibody? How does it compare to something like 1F9 (anti-AMA1) or Rh5/Basigin antibodies? Furthermore (and of more concern) mechanistically, how would these antibodies actually inhibit invasion if they target something that is eventually processed off? I.e. invasion is predicted to proceed without this domain?

Q3. Following this, how do you reconcile the first 197 amino acids comprising the binding region of

P113, when this form alone has a reduced (!) binding to the FL Rh5 (Suppl. Table 1) - shouldn't it be the same as full length P113 (or 1-653 construct?). I.e. Can the authors reconcile Figure 3 and Suppl. Table 1 more completely. In general I found the SPR numbers and different combinations quite confusing!

Q4. "We first showed that the synthetic peptide structurally mimicked RH5Nt by showing it retained the ability to bind P113 with almost identical biophysical binding parameters to RH5FL (Fig. 6a; Supplementary Table 1)" Shouldn't that be binding parameters to Rh5Nt?

Q5. This may be semantics, but in Figure 4 (and references to it in the main text), the 1-653 construct is used and shown to migrate by Native page as a dimer (according to the legend, and possibly size) yet quoted as being used because it doesn't form higher order oligomers. Isn't this also a higher order oligomer (i.e. not a monomer?).

Minor/Discussion Questions:

Q1. Does the processing event at the N terminus allude to the particular role of a class of protease? E.g. Subtilisin or the like? This would be interesting and would suggest the potential for mutagenesis studies (as follow up) changing the residues involved and exploring the functional consequence of Rh5 processing to the invasion process (thereby testing the model in part). I would think this an obvious question that could be mentioned in the discussion at least.

Q2. Is there any conservation/homology between the short linear sequence in the Rh5Nt that is shared among other Rh or other invasion ligand proteins? I.e. could P113 provide a scaffold for more than just Rh5? Again, access to immuno-precipitating antibodies for P113 could not only verify the interaction in vivo but also address this question as well.

Q3. Is there a structural model for the lectin-binding domain of P113 that might allude to the mechanism of interaction with Rh5?

Q4. Is Native page alone enough to conclude tetramerization? Could other techniques (AUC, or SAX) be employed?

Reviewer #2 (Remarks to the Author):

PfRH5 is a leading blood-stage malaria vaccine candidate because anti-PfRH5 antibodies inhibit parasite growth both in vitro and in monkey model, and the interaction with its erythrocyte receptor basigin is essential for invasion. PfRH5 forms complexes with other parasite proteins including CyRPA and RIPR. In this manuscript, the Authors identified P113 as a merozoite surface protein that directly interacts with N-terminal fragment of PfRH5 (PfRH5Nt). Using recombinant proteins and a sensitive protein interaction assay, they analyzed the binding interdependencies of all the other known PfRH5 complex components and concluded that the PfRH5Nt-P113 interaction provides a releasable mechanism for anchoring PfRH5 to the merozoite surface. Finally, they demonstrated that the anti-PfRH5Nt peptide induced growth-inhibitory antibody proposing that PfRH5Nt could contribute to a cost-effective malaria vaccine design. I think this study was conducted by the highly qualified biochemical techniques on the analyses of protein-protein interaction. All the works were carefully designed, clearly presented, and the manuscript is well written. However, I have found missing data of importance as follows:

Major Comments:

1) Since the most important data in this manuscript is PfRH5-P113 interaction in vitro, PfRH5-P113 interaction should be confirmed by the immunoprecipitation of both parasite lysate and culture supernatant (in vivo). The possibility of artifactual interaction between PfRH5-P113 in vitro is still remained.

2) Essential data which confirm the molecular interaction between PfRH5 and P113 must be demonstrated as co-localization by the imaging experiments such as confocal microscopy in the parasite specimen.

3) Supplementary Figure 3

They clearly demonstrated that P113, but not CyRPA, is tethered to the plasma membrane when ectopically expressed in HEK293 cells. The result "CyRPA is not GPI-anchored" is not consistent with the published results by "Reddy KS, et al. Multiprotein complex between the GPI-anchored CyRPA with PfRH5 and PfRipr is crucial for Plasmodium falciparum erythrocyte invasion. Proc Natl Acad Sci U S A 112, 1179-1184 (2015)." This issue should be reconfirmed in the parasite not in HEK293 cells.

Reviewer #3 (Remarks to the Author):

There is much to like about this manuscript, which reports some interesting and important insights into a complex of proteins involved in invasion of RBCs by merozoites. The results are clearly presented and follow a logical order, with carefully planned and thought-out experiments for the most part. The authors report new findings that the invasion ligand RH5 interacts with another protein, P113, on the merozoite surface. They map the binding regions for this interaction, and provide further insights into how a larger invasion complex is assembled, involving CyRPA and RIPR. This is an important finding because RH5 is a promising vaccine candidate, and is thought to be essential for invasion, and understanding these interactions will aid the development of effective vaccines. I believe the authors need to provide a small amount of additional supporting data to support their conclusions, and should address several points for clarification. These are outlined below:

1. I suggest that the title of the MS needs to be changed. The present title reflects their proposed mechanism of releasable tethering, but they provide limited evidence to directly support this conclusion. I would prefer them to use a title that better reflects their key findings

2. The authors use a valuable platform to investigate protein-protein interactions, which they have validated in prior studies. This approach is powerful in revealing the new interactions, and clarifying protein-protein interactions in the invasion complex. A limitation of this approach is the reliance on conclusions drawn only on using recombinant proteins in a simplified model. It would be valuable if the authors could provide some supporting data using native proteins. Of course, interpretations based solely on studies of native proteins can also be misleading, highlighting the value of the approach they have used. Nonetheless, I do believe some complementary data with native proteins would be very valuable here. For example, can they pull-down native RH5 (and other interacting partners) from parasite extracts with recombinant P113, or vice-versa. Can they label P113 in a western blot of the RH5-RIPR-CyRPA complex? Can they show that P113 and RH5 co-localise in microscopy (e.g. super-res microscopy or EM)?

3. It is also unclear what the location of P113 is. The authors describe it as a merozoite surface protein. The earlier report describing P113 (Sanders et al) indicated that localisation is unclear. I am not aware of other data that has defined this. Could the authors comment? It would be valuable to localise P113 to provide a clearer understanding of how the Rh5-P113 complex forms and its mechanism.

4. The authors show that antibodies to RH5-Nt inhibit growth (Fig 6). However, it appears the concentration required was very high (~50% inhibition at 5mg/ml?), which does not make it highly appealing as a vaccine candidate. Some modification of the wording may be appropriate there. Additionally, we need to see confirmation that the anti-RH5-Nt labels RH5 in a western of parasite extract, and does not cross-react with other antigens.

5. Could the authors provide some further clarification of the mechanism of action of the RH5-NT antibodies? How would they be inhibiting, and how does this influence the interpretation of their model? Is the RH5-P113 complex formed prior to schizont rupture - if so, how would antibodies inhibit invasion since the RH5-NT does not interact with basigin. Or are they proposing that the antibodies prevent the P113 interaction; conceivably this could happen before schizont rupture since they add antibodies prior to that time, and others report that antibodies can enter the schizont. Some clarification would be helpful. Also, the assays they use are not actual invasion-inhibition assays, but standard GIAs. It would be good to determine whether the antibodies actually inhibit invasion, and determine the timing of their activity by adding antibodies at defined time points. Would they inhibit if added after schizont rupture (e.g. using video microscopy or isolated merozoites)?

6. The findings do conflict with some earlier reports on roles and interactions with CyRPA and RIPP. Some further discussion to reconcile their findings with the prior data would be helpful for clarification. Mention of the supplementary material investigating whether CyRPA is GPI-anchored I feel would be better somewhere in the results rather than discussion.

Other comments

1. Further references on invasion events and interactions in the introduction (p3) would be helpful

2. The authors describe RH5 as 'conserved' or 'highly conserved'. Could they be more specific?

Polymorphism in RH5 has been reported

3. Please indicate what concentrations of proteins are used in the various interaction assays

4. Worth noting that several studies have shown that RH5 is a target of acquired immunity and antibodies correlate with protection since these findings support RH5 as a vaccine candidate.

Interestingly, it seems that the studies by both Richards et al 2013 and Osier et al 2014 report that antibodies to P113 were associated with protection from malaria. It would be good to mention this as it adds some further interest to P113 as an immune target and vaccine candidate

Point-by-point response to referees' comments.

NCOMMS-16-00495

Galaway *et al.* "The N-terminus of *Plasmodium falciparum* RH5 binds P113 permitting releasable tethering to the merozoite surface."

Please find our point-by-point response to the referees' comments below.

Reviewer #1

The key gap in the paper is in vivo parasite proof that supports the interaction between P113 and Rh5 (and the complex) and provides a native framework when the interaction takes place (i.e. shows its actually important/relevant/real). With several older published studies on P113, I would have hoped that antibodies could be sourced that would allow both the co-immuno-precipitation of the complex (as was done with RIPR and CyRPA) to verify (or at least attempt to verify) that this interaction is indeed confirmed in vivo and just as importantly (or even more) whether the timing of events fits with the model proposed. A core concern is that without such data, i.e. simply relying on the protein-protein work that this could all be an artefact of the in vitro nature of the work (though I am not necessarily suggesting that it is). Memories of aldolase and TRAP-like protein tails come to mind!

The referee raises an important point which was also made by referees 2 and 3. There are several things to consider before attempting to demonstrate that the N-terminus of RH5 and P113 interact using parasite material. Firstly, our model for the function of the RH5-P113 interaction is that it does not form a stable, constitutive complex, but, rather, is a temporally regulated interaction that occurs just fleetingly during the rapid invasion process, which takes only a few seconds. We know from the work of others^{1, 2, 3, 4} (and see discussion later in this rebuttal) that prior to invasion, P113 and RH5 are segregated within the intact merozoite and thereby purposefully prevented from interacting. And secondly, it has been demonstrated by others using co-purification experiments from parasite culture supernatants that the vast majority of RH5 is found within a "post-invasion" RH5/CyRPA/RIPR complex⁴, which, as we have shown in our paper, is no longer able to bind P113 (see Fig. 5f). Finally, while we have raised polyclonal antibodies to P113, RH5FL and RH5Nt, which we could use for co-purification experiments, we know that the epitopes recognised by these antibodies overlap with the P113-RH5 binding sites on both proteins (see Fig. 6c). We therefore believe that adding these antibodies would be unsuitable for co-immunoprecipitation experiments because they would rapidly displace their interaction partners making the interaction difficult, if not impossible, to detect. Taking these factors into account, we have decided to follow the suggestion made by referee 3 to address this point by separately immobilising RH5Ct and RH5Nt on agarose beads, and incubate these with a parasite lysate to determine if native P113 could be specifically purified with RH5Nt (but not RH5Ct which is used as a control) by using anti-P113 Western blotting.

At first, we solubilised the parasite material using the detergent Triton X-100 since P113 was previously identified as a component of detergent-resistant membranes that required a "strong" non-ionic detergent for solubilisation². Triton X-100 did indeed effectively solubilise P113 (see Fig. R1A), and while we could detect a band of the appropriate size in the RH5Nt-coated but not RH5Ct-coated beads, it was not entirely convincing, nor reproducible (Fig.

R1A). We postulated that the Triton X-100 detergent might inhibit the P113-RH5 interaction. To test this, we titrated increasing concentrations of Triton X-100 into our P113-RH5 binding assay that uses highly avid pentameric recombinant proteins (the AVEXIS assay). Indeed, we observed a dose-dependent inhibition of the RH5-P113 interaction with Triton X-100 (Fig. R1B). We therefore switched to using a slightly gentler non-ionic detergent, NP40, which we found did not inhibit the P113-RH5 interaction. NP40 was not as effective at solubilising P113, but by using mechanical disruption (vortexing and repeated pipetting) we could get sufficient P113 into solution (Fig. R1C). Under these conditions, we were able to convincingly and reproducibly show purification of P113 from parasite lysates with RH5Nt, but not with the RH5Ct control (Fig. R1C). This is an important result and we have added this experiment to the revised manuscript as a new Figure 2c including the following sentence in the result section, and updated the methods accordingly.

“We further demonstrated this interaction by showing that RH5Nt but not control RH5Ct-coated beads could purify native P113 from parasite culture lysates (Fig. 2c).”

Figure R1: Biochemical purification of P113 from *P. falciparum* blood stage cultures using RH5Nt, but not RH5Ct. 3D7 parasites at mixed trophozoite and schizont stages were isolated on Percoll gradients, saponin treated to remove erythrocyte material and solubilised with 1% Triton X-100. Streptavidin-coated agarose beads were coated with biotinylated RH5Ct-Cd4d3+4bio or RH5Nt- Cd4d3+4bio and were incubated with the parasite lysate, washed, and the eluate analysed by SDS-PAGE/Western blot with anti-P113 antibodies. **(A)** A band corresponding to full length P113 is present in the total parasite lysate “Triton lysate” at the expected full length mass (113 kDa) and several smaller (possibly cleavage) fragments. Beads coated in RH5Nt, but not RH5Ct purified a band corresponding to full length P113, although this was neither entirely convincing nor reliably repeatable. **(B)** Increasing concentrations of Triton X-100 inhibited the RH5-P113 interaction. The AVEXIS assay was used to detect the RH5-P113 interaction using P113 as the pentameric prey and RH5Nt as the biotinylated bait. Increasing concentrations of Triton X-100 inhibited the interaction. “-” represents a negative control. **(C)** Biochemical purification of P113 from NP40-solubilised *P. falciparum* blood stage cultures using RH5Nt, but not RH5Ct. Co-purification of P113 with RH5Nt, as in **(A)**, but from parasite lysates solubilised in NP40 and not Triton X-100. The bands observed at ~50kDa (RH5Ct) and ~30kDa (RH5Nt) correspond to the RH5Ct-Cd4d3+4bio or RH5Nt- Cd4d3+4bio proteins. They are detected because they contain the Cd4d3+4 tag which was also present on the P113-Cd4d3+4 immunogen used to raise the anti-P113 antibody. This result with the NP40 detergent was independently replicated, and the results of a different experiment are shown in the new Figure 2c, where we have cropped the gel to remove the bands corresponding to the RH5Ct-Cd4d3+4bio or RH5Nt- Cd4d3+4bio proteins to avoid confusion.

For example, the timing of the interaction (and spatial localisation of proteins involved) would appear to me to be the key to how this would or would not work. E.g. How does the complex with RIPR NOT come together? If RIPR is a micronemal protein it would be secreted at the same time or even prior according to current understanding so it would be around. Thus complex formation must be regulated by the unknown protease - but how? Similarly, when is Rh5 actually processed? Some proteins are pre-processed before they are even released. If it is already in a processed form in the schizont, then it might be that P113 binding is already compromised before invasion even begins - not so good for the model and not so good for a vaccine either.

Since the other referees have raised similar points, it is worth discussing what one might expect in terms of the co-localisation of the proteins and the temporal ordering of interactions that form part of the RH5 complex within the parasite. It has been established using immunoelectron microscopy that many parasite ligands that are involved in invasion are located within discrete intracellular organelles (e.g. micronemes or rhoptries), and are secreted in a localised manner at the apical tip of the merozoite just at the precise moment of invasion. In the case of the proteins that are already established to be part of the RH5 invasion complex (RH5, CyRPA and RIPR) they have been reported to be localised within different organelles in the free merozoite: RH5 is located within the rhoptries, whereas CyRPA and RIPR are found within the micronemes^{1, 4, 5}. We therefore already know that these well-established members of the RH5 complex do not co-localise in the pre-invasion merozoite, but expect that they briefly interact during invasion and remain associated as a complex in the parasite culture supernatant post invasion, which is how this complex was discovered. The fact that we normally observe these proteins segregated into different subcellular organelles is therefore an important aspect of the regulatory mechanism by which the parasite ensures these proteins only interact at the correct time and place. Using a state-of-the-art imaging technique stimulated emission depletion (STED) super resolution imaging Volz *et al.* in their recent paper “Essential role of the PfRh5/PfRipr/CyRPA complex during *Plasmodium falciparum* invasion of erythrocytes” managed to show some colocalisation between RH5, CyRPA and RIPR; however, even using this detailed approach, it was surprising that these proteins showed little co-localisation, leading the authors to remark:

“A surprising result was the lack of full colocalization of PfRh5 with PfRipr or CyRPA over the merozoite, suggesting not all the PfRh5 pool forms a complex. Indeed, colocalization of PfRh5, PfRipr, and CyRPA at the apical end of merozoites as it interacts with the erythrocyte membrane suggests the complex forms at the site of function for invasion.”

This suggests that only a fraction of each protein is at any one time part of the complex and/or that the interaction occurs within a very tight time window – perhaps in just a few seconds or less. It is one of the major challenges in the field to develop methods that would enable the visualisation of these protein interactions occurring in real time during the rapid invasion process.

An important contribution of this manuscript is the identification of P113 as a new member of the RH5 complex. While it has been previously established that P113 is a GPI-anchored protein and therefore expected to be localised on the surface of merozoites^{2, 6}, this has not been unequivocally demonstrated and published. We therefore used our anti-P113 antisera and immunocytochemistry to determine the subcellular localisation of P113. By using fluorescent immunocytochemistry and confocal microscopy, we could show that the anti-P113 antibodies stained the parasites in a pattern that is consistent with it being present in the plasma membrane. There was a good correlation with a well characterised Inner

Membrane Complex (IMC) marker, MTIP⁷, which lies immediately under the plasma membrane in late schizonts (Fig. R2). These data have been added to the revised manuscript by including the following sentence and a new supplementary figure (Supplementary Figure 1).

“Antibodies raised against P113 stained the periphery of blood stage *P. falciparum* parasites, immediately adjacent to the inner membrane complex marker MTIP⁷, confirming that P113 is located within the parasite blood stage plasma membrane (Supplementary Fig. 1).”

Figure R2: Antibodies against P113 localise to the parasite plasma membrane. Fixed and permeabilised blood stage parasites were probed with anti-P113 antibodies (red), the membrane marker MTIP (green) and nucleic acid stained with DAPI (blue). P113 staining is consistent with it being localised in the parasite plasma membrane. Scale bar represents 2 μ m.

Specific Questions:

Q1. What is the mechanism/basis for 113 binding to the full length of Rh5 with a 10 fold higher affinity than the processed N terminus? Presumably there must be some structural change or masking in the processed form? Or is the binding site partly covered by both regions N and C terminal?

This is an interesting question and one that we were able to investigate in more detail by performing a comparative thermodynamics analysis using surface plasmon resonance of the RH5Nt and RH5FL binding to P113. This essentially involves quantifying and comparing the enthalpic and entropic contributions to the energy released upon P113 binding for both RH5Nt and RH5FL.

Our experiments show that the P113 interaction with RH5 appears to be entropically driven with a favourable ΔS value under equilibrium binding conditions (Figure R2). The magnitude of ΔS is greater for RH5FL than RH5Nt, which causes a lower free energy minimum for RH5FL compared to RH5Nt and therefore explains its stronger binding affinity for P113. It is often found that protein-protein interactions at cell surfaces are entropically driven⁸. The interpretation of these data are that the N-terminus of RH5 may be partially ordered (or in some way structurally constrained) when in proximity to the C-terminus of RH5, and that the interaction with P113 would release RH5Nt from this restraint thereby causing an overall increase in disorder and therefore a more favourable entropic contribution to the binding. The RH5Nt fragment protein may already be relatively disordered and so has a weaker entropic driver.

Figure R2: Thermodynamic surface plasmon resonance analysis of P113 interaction with RH5 FL and RH5Nt. P113 Y1-N653 was injected over immobilised RH5FL (500 RU) and RH5Nt (240 RU) (150RU of Cd4 control was immobilised in the reference flow cell) at different temperatures (13-37°C) and concentrations (2 fold serial dilution of 2 to 0.0625 μ M). Kinetic association and dissociation values were calculated. From these were plotted a van't Hoff plot (A) and Eyring plots (B and C) from which the entropy, enthalpy and free energy components of the interactions could be calculated (D). Circles represent RH5FL and squares RH5Nt.

We have included these data as a supplementary figure (Supplementary Fig. 2) together with a statement which summarises these findings in the results section of our revised manuscript:

“We also performed a thermodynamics analysis of the RH5-P113 interaction using SPR which showed that the interaction was entropically driven and that the entropic contribution was higher for RH5FL compared to RH5Nt providing an explanation for the 10-fold higher affinity of RH5FL for P113 in comparison to RH5Nt (Supplementary Fig. 2). This suggests that the N-terminus of RH5 may be partially ordered or in some way structurally constrained when in proximity to the C-terminus of RH5, and that the interaction with P113 would release RH5Nt from this restraint thereby causing an overall increase in disorder and therefore a more favourable entropic contribution upon P113 binding.”

Q2. The activity of antibodies elicited by the synthetic peptide of Rh5 is interesting. However, the activity appears to be quite weak (in the 1-2+ (may be even 10) mg range). This would not appear to me to be potent enough to be a serious contender for an inhibitory antibody? How does it compare to something like 1F9 (anti-AMA1) or Rh5/Basigin antibodies? Furthermore (and of more concern) mechanistically, how would these antibodies actually inhibit invasion if they target something that is eventually processed off? I.e. invasion is predicted to proceed without this domain?

The referee makes the point that the potency of the polyclonal antibodies targeting a peptide corresponding to just the N-terminal region of RH5 appears relatively weak when used in parasite invasion assays, and asks how their potency compares to other antibodies that inhibit invasion. It is certainly our experience that we need to use much higher concentrations of anti-RH5 antibodies to prevent invasion in comparison to antibodies that target the basigin receptor. We discuss this in detail in our recent paper “Basigin is a druggable target for host-oriented antimalarial interventions” Zenonos ZA *et al.*⁹, pointing out that the IC_{50} of monoclonal antibodies to the basigin receptor is typically $\sim 1\mu$ g/mL whereas monoclonal antibodies targeting RH5 have IC_{50} s in the range of 15 to 100 μ g/mL. We and others believe that the likely reason for this is the accessibility of the respective proteins: the basigin receptor is constantly exposed to antibodies on the surface of the erythrocyte enabling sufficient time for the antibody to achieve binding equilibrium. By contrast, RH5 is thought to be only fleetingly exposed since it is secreted by the parasite at the moment of invasion, and is therefore likely to require higher antibody concentrations.

It is very difficult to compare the potencies of monoclonal (e.g. 1F9) and polyclonal antibodies since one would expect that only a fraction of the antibodies in polyclonal antisera raised to a particular antigen will target epitopes that have protective effects. Monoclonals targeting the same antigen could, however, be either very potent or have no activity at all depending on the epitope recognised. While our paper provides a possible mechanistic explanation, others have shown that a monoclonal antibody that targets the N-terminal region of RH5 can have very potent ($IC_{50} \ll 100 \mu\text{g/mL}$) parasite invasion blocking effects (see: "A malaria vaccine candidate based on an epitope of the *Plasmodium falciparum* RH5 protein" Ord *et al.* ³). Vaccines elicit a polyclonal response and so quantitatively comparing their inhibitory activity to monoclonal antibodies isn't always appropriate in terms of ranking their candidacy for a vaccine target. To answer the referee's question directly, then it is our experience that the polyclonal antisera raised against the full length RH5 protein in rabbit generally has lower IC_{50} s than the antisera raised against the 116 amino acids that make up the N-terminal region. This is perhaps unsurprising since the antisera raised against full length RH5 protein will include antibodies that target all epitopes on the RH5 protein, and not just those in the N-terminal region. Note that not all monoclonal antibodies to RH5 are inhibitory (see "Neutralization of *Plasmodium falciparum* merozoites by antibodies against PfRH5" Douglas AD *et al.* ¹⁰) demonstrating that there are non-protective epitopes on RH5. A current challenge in vaccinology is to isolate the protective regions of an antigen so as to focus the polyclonal response on these regions. We believe that the contribution we have made in this study is to identify a possible protective region of RH5 and (importantly) provide a mechanistic explanation for its functional role. Critically, because this region of RH5 can be structurally mimicked by chemically synthesized peptides, this raises the possibility that the protective region could be narrowed further which would likely increase the potency of the vaccine. It is our future research plan to further refine these regions on RH5 to see if we can elicit a more potent polyclonal antibody response both by using shorter peptide antigens and (particularly) with collaborators obtain a co-crystal structure of the RH5Nt-P113 complex to determine the structural characteristics of the interaction interface to inform vaccine design.

The referee also asks how antibodies can inhibit invasion if they target something that is eventually processed. There is a large and growing body of evidence from several groups that the RH5-basigin interaction is essential for parasite invasion of erythrocytes, and that N-terminal processing is required for RH5 function. Our manuscript describes a mechanistic explanation for the processing event by identifying a binding partner for the RH5 N-terminal region. We believe that antibodies that target the RH5Nt region are therefore likely to interfere with this processing event, interfere with RH5 function and consequently inhibit invasion.

Q3. Following this, how do you reconcile the first 197 amino acids comprising the binding region of P113, when this form alone has a reduced (!) binding to the FL Rh5 (Suppl. Table 1) - shouldn't it be the same as full length P113 (or 1-653 construct?). I.e. Can the authors reconcile Figure 3 and Suppl. Table 1 more completely. In general I found the SPR numbers and different combinations quite confusing!

Thanks to the reviewer for pointing this out. We have looked closely at the surface plasmon resonance data and especially how well the experimental data fit a theoretical model of a typical 1:1 interaction. Following this reanalysis we have improved these fits which has resulted in updated values as follows: $k_a = 2.84 \pm 0.05 \times 10^5 \text{ M}^{-1}\text{s}^{-1}$, $k_d = 0.307 \pm 0.002 \text{ s}^{-1}$, $K_D \text{ calc.} = 1.1 \mu\text{M}$ and $t_{1/2} = 2.3 \text{ s}$ and we have updated Supplementary Table 1 accordingly.

We believe that perhaps the most satisfactory way of addressing the question of how P113 and the N-terminal region of RH5 interact would be to solve the structure of the P113-RH5 complex, and we have initiated a collaboration with a crystallographer, Professor Matthew Higgins at the University of Oxford to address this in detail; however, we think that such structural studies are clearly beyond the scope of this manuscript.

Q4. "We first showed that the synthetic peptide structurally mimicked RH5Nt by showing it retained the ability to bind P113 with almost identical biophysical binding parameters to RH5FL (Fig. 6a; Supplementary Table 1)" Shouldn't that be binding parameters to Rh5Nt?

Yes, the referee is quite right – now corrected. Thanks.

Q5. This may be semantics, but in Figure 4 (and references to it in the main text), the 1-653 construct is used and shown to migrate by Native page as a dimer (according to the legend, and possibly size) yet quoted as being used because it doesn't form higher order oligomers. Isn't this also a higher order oligomer (i.e. not a monomer?).

The referee is quite correct to point this out and we agree that we have not been precise enough in our interpretation and description of these data. While it is clear that the mass of the truncated form of P113 that lacks the coiled coil region is far smaller than the entire ectodomain, demonstrating that it is no longer able to form higher-order oligomeric forms, it is not certain that all the molecular species are monomeric. The native PAGE does show two molecular species for the truncated form and so we have modified our manuscript by being more precise in our description of these data in the figure legend. The surface plasmon resonance data using the truncated P113 as an analyte clearly show a 1:1 binding behaviour which may suggest that one of the two forms might not be able to bind RH5. As we have previously mentioned, we believe that the best and most conclusive way of characterising this interaction in more detail will be to solve the co-crystal structure of P113 with the N-terminal region of RH5 and as noted above, we have initiated a collaboration with Professor Matthew Higgins at the University of Oxford to address this directly.

Minor/Discussion Questions:

Q1. Does the processing event at the N terminus allude to the particular role of a class of protease? E.g. Subtilisin or the like? This would be interesting and would suggest the potential for mutagenesis studies (as follow up) changing the residues involved and exploring the functional consequence of Rh5 processing to the invasion process (thereby testing the model in part). I would think this an obvious question that could be mentioned in the discussion at least.

During this study we did indeed investigate the ability of different protease inhibitors to prevent the RH5 processing when expressed in HEK293 cells. We observed that several protease inhibitors were able to prevent RH5 processing, with aprotinin being particularly effective at concentrations between 1 and 10 µg/mL (Figure R3). As we describe in the methods, we subsequently used aprotinin in our RH5 expression cultures to ensure that we obtained full length RH5 protein for our biochemical studies.

Figure R3: Protease inhibitors and particularly aprotinin prevented RH5 processing in HEK293 cells. HEK293 cells were grown in 1% fetal calf serum and transfected with a plasmid encoding the full-length *P. falciparum* RH5 protein in the presence of named protease inhibitors and the supernatant resolved by SDS-PAGE, blotted and RH5 detected with a polyclonal antibody raised to the full length RH5 protein. Most cleavage events can be prevented with protease inhibitors in the culture medium, although aprotinin was particularly effective, even at low concentrations.

Aprotinin is known to inhibit serine proteases which do, as the referee suggests, include subtilisins. RH5 was not a predicted or experimentally validated substrate of SUB1 (perhaps the most well-known of *P. falciparum* subtilisins) as described by de Monerri *et al.* “Global identification of multiple substrates for *Plasmodium falciparum* SUB1, an essential malarial processing protease”¹¹. As the referee suggests though, this is an interesting finding and should be highlighted since it may provide impetus for further studies. We have therefore added the following sentences to the discussion:

“The identification of the protease/s responsible for RH5 cleavage will be an important step in understanding the function of the RH5 invasion complex. Whilst the observation that aprotinin can reduce RH5 processing suggests the involvement of a serine protease, RH5 is not a predicted or experimentally validated substrate of the well-characterised parasite blood stage serine proteases, the subtilisins including SUB1.”

Q2. Is there any conservation/homology between the short linear sequence in the Rh5Nt that is shared among other Rh or other invasion ligand proteins? I.e. could P113 provide a scaffold for more than just Rh5? Again, access to immunoprecipitating antibodies for P113 could not only verify the interaction *in vivo* but also address this question as well.

A BLAST search against the *P. falciparum* predicted proteome with the 19 amino acids that contain the P113 binding site identified a single but weak hit with another parasite protein – the alignment is shown below for information.

```

RH5Nt  4  ENNLALLPIKSTEEK  19
      ENN  LLP+K T E K
RER1   88  ENNGLLLPMKQTHETK  103
    
```

The protein is known as RER1 or “retrieval receptor for endoplasmic reticulum membrane proteins” (PF3D7_0903100), which is an uncharacterised protein but named as such because it contains a domain that is found in proteins that are involved in the retrieval of proteins back to the endoplasmic reticulum from the early Golgi compartment. RER1 is an integral membrane protein containing three predicted transmembrane-spanning regions. The amino acids 88 to 103 that contain the homology to RH5Nt are located just after the first transmembrane helix which is predicted to be cytoplasmic (Figure R4).

Figure R4. RER1 is unlikely to interact with P113. Output from the transmembrane predictor TMHMM for *P. falciparum* protein RER1 shows that the region with weak homologies to the P113 binding site on RH5 (amino acids 88 to 103) lies just after the first predicted transmembrane domain and is therefore predicted to be intracellular.

Since we submitted this manuscript, P113 has been identified as a peripheral component of the PTEX protein translocation machinery during the intraerythrocytic stage of the parasite lifecycle (see Elsworth *et al.* “Proteomic analysis reveals novel proteins associated with the *Plasmodium* protein exporter PTEX and a loss of complex stability upon truncation of the core PTEX component, PTEX150” Cellular Microbiology 2016 ⁶). Using gentle isolation conditions and crosslinking approaches, peptides corresponding to P113 were present in PTEX complex purifications. Because this study also concluded that P113 is tethered to the parasite membrane by a GPI anchor, this suggested P113 might act as a bridge or spacer molecule between the parasite membrane and the membrane forming the parasitophorous vacuole. This proposed function of P113, at a different step of the lifecycle, would not be mutually incompatible with a role on the merozoite surface during invasion. This is an interesting finding and we have added a clause which mentions this in the discussion and cites this study.

“and more recently, P113 has been shown to be a peripheral member of the PTEX translocation machinery which is conserved across *Plasmodium* spp.⁶.”

Q3. Is there a structural model for the lectin-binding domain of P113 that might allude to the mechanism of interaction with Rh5?

Many thanks to the referee for pointing this out, but despite a concerted effort, we were unable to find any protein domain prediction programme that identified a lectin domain in P113. We have searched several databases including Pfam (<http://pfam.xfam.org/>), Interpro (<https://www.ebi.ac.uk/interpro/>), SMART (<http://smart.embl-heidelberg.de/>), TIGRFAM (<http://www.jcvi.org/cgi-bin/tigrfams/index.cgi>), Gene3D (<http://gene3d.biochem.ucl.ac.uk/Gene3D/>), Superfamily (<http://supfam.org/SUPERFAMILY/>) or PIRSF (<http://pir.georgetown.edu/pirwww/dbinfo/pirsf.shtml>). We don't dispute the possibility that P113 might contain a lectin domain, but the sequence matches with lectin domains that have been characterised so far don't appear to be very strong, and so we would not have a great deal of confidence interpreting anything derived from a structural model. We'd like to reassure the referee that this question is being addressed through structural studies – it is quite likely that structural homologies will be more obvious than sequence comparisons.

Q4. Is Native page alone enough to conclude tetramerization? Could other techniques (AUC, or SAX) be employed?

The referee is correct to point out that we shouldn't rely on native PAGE results alone to conclude the oligomeric status of the entire ectodomain of P113. From the gel filtration and native PAGE results on the entire ectodomain of P113 and the drastic shift in the elution profile and loss of binding avidity when the predicted coiled coil region is removed, we are confident that P113 does form higher order oligomers; however, as the referee suggests, these are not high resolution techniques, and so we can't be fully confident of the stoichiometry. As suggested, analytical ultracentrifugation (AUC) and small angle X-ray scattering (SAX) would lend support to these findings, although because of the very large size of the oligomers (>500 kDa), we're not confident that either would provide a definitive answer.

We have reviewed our wording around this result and, being mindful of the limitations of the techniques we have used as discussed above, we believe we have been appropriately cautious in our interpretation, referring to native P113 forming an oligomer or multimer in the main text and in the Figure 4 and 7 legends. We do suggest that P113 might form a tetramer with the wording "consistent with the formation of tetrameric complexes", "P113 clusters, probably as tetramers", and reference a study where a similar coiled coil region in MSP3 was thought to form tetramers. Therefore, we're inclined to retain the cautious wording around these data and their interpretation, but if the referee objects and is of the opinion that we have over-interpreted the data, then we'd be happy to revise the wording accordingly.

Reviewer #2

Major Comments:

1) Since the most important data in this manuscript is PfrH5-P113 interaction in vitro, PfrH5-P113 interaction should be confirmed by the immunoprecipitation of both parasite lysate and culture supernatant (in vivo). The possibility of artifactual interaction between PfrH5-P113 in vitro is still remained.

The referee raises an important point and we refer them to the discussion and results we have given to address the same question from referee 1. In brief, we were able to

reproducibly purify native P113 from parasite lysates using agarose beads coated in RH5Nt, but not with the control RH5Ct. Note that as we have discussed earlier in this rebuttal, we would not expect to co-purify P113 as part of the RH5 complex in parasite culture supernatants because RH5 in this “post-invasion” complex is already processed and lacks the P113-binding N-terminal region.

2) *Essential data which confirm the molecular interaction between PFRH5 and P113 must be demonstrated as co-localization by the imaging experiments such as confocal microscopy in the parasite specimen.*

Again, the referee raises an important point and we refer them to the answers and discussion we have given to address the same question from referee 1. In brief, the fact that RH5 is located in the rhoptries and CyRPA/RIPR in the micronemes suggests that the formation of the RH5 complex is regulated, and one important component of this regulation is to segregate the proteins within different subcellular organelles so that they are not normally co-localised within the merozoite prior to invasion. We expect that RH5 and P113 would be colocalised for only a fleeting moment during the invasion process before being released either by the processing of RH5 to remove RH5Nt, or the recruitment of RIPR to the complex which, as we have shown is then incompatible with P113 binding. Similar findings were reported for the co-localisation of the other members of the RH5 complex at the point of invasion in a recent paper from Volz *et al.* in their recent paper “Essential role of the PFRh5/PFRipr/CyRPA complex during *Plasmodium falciparum* invasion of erythrocytes” – here they used STED super resolution microscopy and reported that the established members of the RH5 complex: RH5, CyRPA and RIPR showed only limited colocalisation even when individual invasion events were selected and analysed. One important question that the referee raises is the subcellular localisation of P113 and we have now used antibodies to P113 together with fluorescent immunocytochemistry and confocal microscopy to show that P113 is located on the parasite plasma membrane (Fig. R2).

3) *Supplementary Figure 3*

They clearly demonstrated that P113, but not CyRPA, is tethered to the plasma membrane when ectopically expressed in HEK293 cells. The result "CyRPA is not GPI-anchored" is not consistent with the published results by "Reddy KS, et al. Multiprotein complex between the GPI-anchored CyRPA with PFRH5 and PFRipr is crucial for Plasmodium falciparum erythrocyte invasion. Proc Natl Acad Sci U S A 112, 1179-1184 (2015)." This issue should be reconfirmed in the parasite not in HEK293 cells.

We fully agree with the referee that this is a crucial point and one which has important mechanistic implications for understanding how RH5 functions. During the submission of this manuscript, we became aware of studies from the laboratory of Professor Alan Cowman who has investigated the question of whether CyRPA is GPI-anchored in detail. Their results are now published (see: Volz *et al.* in their recent paper “Essential role of the PFRh5/PFRipr/CyRPA complex during *Plasmodium falciparum* invasion of erythrocytes”) and their results are consistent with ours and find no evidence that CyRPA is a GPI-anchored protein, and in fact directly show that CyRPA is not GPI-anchored.

As suggested by referee 3, we have moved the description of these results from the discussion to the results section and have cited the manuscript referred to above.

Reviewer #3

1. I suggest that the title of the MS needs to be changed. The present title reflects their proposed mechanism of releasable tethering, but they provide limited evidence to directly support this conclusion. I would prefer them to use a title that better reflects their key findings

We agree with the referee and have changed the title to “P113 is a merozoite surface protein that binds the N-terminus of *Plasmodium falciparum* RH5.”

2. The authors use a valuable platform to investigate protein-protein interactions, which they have validated in prior studies. This approach is powerful in revealing the new interactions, and clarifying protein-protein interactions in the invasion complex. A limitation of this approach is the reliance on conclusions drawn only on using recombinant proteins in a simplified model. It would be valuable if the authors could provide some supporting data using native proteins. Of course, interpretations based solely on studies of native proteins can also be misleading, highlighting the value of the approach they have used. Nonetheless, I do believe some complementary data with native proteins would be very valuable here. For example, can they pull-down native RH5 (and other interacting partners) from parasite extracts with recombinant P113, or vice-versa. Can they label P113 in a western blot of the RH5-RIPR-CyRPA complex? Can they show that P113 and RH5 co-localise in microscopy (e.g. super-res microscopy or EM)?

The referee has raised the same important point made by both referee 1 and 2 and helpfully provides suggestions (that we have used) to answer this point experimentally. We have shown earlier in this rebuttal how we were able to purify native P113 using RH5Nt but not the control RH5Ct protein. Please also see our answers to referee 1 and 2 above that addresses the points about co-localisation of RH5 and P113.

3. It is also unclear what the location of P113 is. The authors describe it as a merozoite surface protein. The earlier report describing P113 (Sanders et al) indicated that localisation is unclear. I am not aware of other data that has defined this. Could the authors comment? It would be valuable to localise P113 to provide a clearer understanding of how the Rh5-P113 complex forms and its mechanism.

We agree with the referee that this is an important point and one that was raised by both the other referees. We have used anti-P113 antibodies to show that P113 is localised to the parasite plasma membrane (Fig. R2), and have included these data as a new supplementary figure in our revised manuscript.

4. The authors show that antibodies to RH5-Nt inhibit growth (Fig 6). However, it appears the concentration required was very high (~50% inhibition at 5mg/ml?), which does not make it highly appealing as a vaccine candidate. Some modification of the wording may be appropriate there. Additionally, we need to see confirmation that the anti-RH5-Nt labels RH5 in a western of parasite extract, and does not cross-react with other antigens.

We refer the referee to the answer we have given to referee 1 Q2 above in regard to the potency of the anti-RH5Nt antibodies.

To demonstrate the specificity of the anti-RH5Nt polyclonal antibodies elicited against the chemically synthesized peptide, we used it in a Western blot of parasite supernatants which should contain processed RH5 (RH5Ct and RH5Nt) fragments. We were able to detect a clear band at the expected size of approximately 14kDa (Fig. R5).

Figure R5: *Plasmodium falciparum* spent culture supernatant probed with anti-pRH5Nt polyclonal antibodies. *P. falciparum*-infected erythrocytes that had been Percoll purified were seeded into fresh RPMI without erythrocytes and allowed to rupture over 48 hours. The supernatant was collected and probed by Western blot with polyclonal antibodies raised against the synthetic RH5Nt.

5. Could the authors provide some further clarification of the mechanism of action of the RH5-NT antibodies? How would they be inhibiting, and how does this influence the interpretation of their model? Is the RH5-P113 complex formed prior to schizont rupture - if so, how would antibodies inhibit invasion since the RH5-NT does not interact with basigin. Or are they proposing that the antibodies prevent the P113 interaction; conceivably this could happen before schizont rupture since they add antibodies prior to that time, and others report that antibodies can enter the schizont. Some clarification would be helpful. Also, the assays they use are not actual invasion-inhibition assays, but standard GIAs. It would be good to determine whether the antibodies actually inhibit invasion, and determine the timing of their activity by adding antibodies at defined time points. Would they inhibit if added after schizont rupture (e.g. using video microscopy or isolated merozoites)?

These are again important questions raised by the referee. We firstly refer the referee to the answer we have given to referee 1 Q2 above. In brief, we believe that the components of the RH5 complex are spatially segregated within the merozoite prior to invasion: RH5 is located in the rhoptries, CyRPA and RIPP in the micronemes, and (as we have shown in this rebuttal) P113 at the merozoite plasma membrane. During the rapid invasion process (which takes just a matter of a few seconds), regulated secretion of the micronemes and rhoptries enables these proteins to interact and form a transient complex that is necessary for invasion to occur. Post invasion, we know from the work of others that RH5, CyRPA and RIPP form a stable complex (which as we show in Fig. 5f cannot bind P113) in parasite culture supernatants that can be co-purified^{4, 5}. The model that we propose in Figure 7 is consistent with the data we currently have, and suggests that the P113-RH5 complex does not form prior to schizont rupture, but after rhoptry secretion during invasion. Our data are consistent with the interpretation that the antibodies to RH5Nt disrupt invasion by preventing the formation of the P113-RH5 complex at the merozoite surface. The anti-RH5Nt antibodies may also prevent the RH5 processing event.

As the referee suggests, we should be able to test if the anti-RH5Nt antibodies are inhibiting invasion since they should prevent the formation of new ring stages which depend on invasion events but not inhibit normal blood stage progression (the formation of trophozoites and schizonts from ring stages). We therefore added anti-RH5Nt antibodies and quantified the number of parasites at the ring, trophozoite and schizont stages over a 48 hour cycle

and compared them to a negative (media) and positive (anti-basigin monoclonal antibody). In the positive control wells containing only media, (Fig. R6a) a large increase in ring stage parasites was observed between 40 to 48 hrs as re-invasion occurs. The number of re-invasion events is reduced by the anti-pRH5Nt antibodies compared to media alone (compare Fig. R6a and Fig. R6b at 48 hours, and their ratio of rings at 48 hrs to schizonts at 40hrs in Fig. R6d) suggesting that anti-RH5Nt antibodies inhibit invasion. Note that the blood stages progress as expected in the presence of anti-RH5Nt antibodies (rings progress to trophozoites and then schizonts) demonstrating that the antibodies are not generally toxic to the parasites, and the ratio of schizonts at 40 to 48 hours is not significantly affected, suggesting no block in schizonts development or in egress. Note that the very strong invasion inhibitory effect of the anti-basigin monoclonal antibodies is consistent with our previously published work¹². We have modified our manuscript accordingly by adding these data as an additional supplementary figure (Supplementary Fig. 6) together with a statement in the appropriate results section

“and (anti-Rh5Nt antibodies) specifically affected schizont to ring stage progression demonstrating that they inhibited erythrocyte invasion (Fig. 6e and Supplementary Fig. 6)”.

Figure R6: Anti-RH5Nt antibodies inhibit parasite growth by preventing invasion. A time course of blood stage development was performed in the presence of polyclonal antibodies against RH5Nt (pRH5Nt), media alone or an anti-basigin mAb. Smears were made at intervals over 48 hrs from blood stage cultures of *P. falciparum* 3D7 in RPMI alone (a), with polyclonal antibodies against pRH5Nt in RPMI at 4 mg/ml (b) or basigin monoclonal antibodies in RPMI at 10 μ g/ml (c). The number of rings, trophozoites and schizonts at each time point (0, 24, 40 and 48 hours) were counted after being smeared, fixed and stained with Giemsa and 2000 erythrocytes were examined by light microscopy. (d) The ratio of rings at 48 hrs to schizonts at 40 hrs is shown in dark grey and the ratio of schizonts at 48 and 40 hrs is shown in light grey. The bars represent means ($n = 3$) and error bars 95% confidence intervals.

6. *The findings do conflict with some earlier reports on roles and interactions with CyRPA and RIPR. Some further discussion to reconcile their findings with the prior data would be helpful for clarification. Mention of the supplementary material investigating whether CyRPA is GPI-anchored I feel would be better somewhere in the results rather than discussion.*

We agree with the referee that this is an important experiment which can help clarify the mechanism by which RH5 functions. As suggested, we have moved the description of this experiment from the discussion to the end of the results section entitled “The RH5-CyRPA-RIPR complex can interact with basigin, but not P113”; consequently, Supplementary Figures 2 and 3 have been reordered. Please also note our reply to referee 2 above where we point out that another group has taken a different approach to answering this question and have reached the same conclusion that CyRPA is not GPI-anchored – we have added the citation to this study in this section.

Other comments

1. *Further references on invasion events and interactions in the introduction (p3) would be helpful*

Agreed. We have now added citations to several recent review and primary research articles that describe the molecular basis of erythrocyte invasion by *Plasmodium* parasites into the introduction.

2. *The authors describe RH5 as 'conserved' or 'highly conserved'. Could they be more specific? Polymorphism in RH5 has been reported*

Yes, non-synonymous polymorphisms in RH5 have been reported in population-based parasite genome sequencing studies (e.g. Manske, M. et al. “Analysis of *Plasmodium falciparum* diversity in natural infections by deep sequencing.”¹³) which described twelve protein coding polymorphisms in RH5, of which only five were reported at a frequency of over 0.05% in parasites sequenced from Africa, Asia and Papua New Guinea. These polymorphisms were documented in Bustamante, L. Y. et al. A full-length recombinant *Plasmodium falciparum* PfRH5 protein induces inhibitory antibodies that are effective across common PfRH5 genetic variants¹⁴. Only one of the twelve polymorphisms (N88D) is located in the N-terminal region of RH5, and this was reported at a frequency of 1% in parasites obtained from Africa, and has not yet been detected in isolates from other regions. We have now added this additional explanatory information into the introduction of the manuscript:

“Despite this, RH5Nt is particularly well conserved with just a single non-synonymous polymorphism described at very low frequency (<1%) in African isolates”

3. *Please indicate what concentrations of proteins are used in the various interaction assays*

We have now explicitly added the concentrations of the baits and preys used in the AVEXIS screens to the methods. We have also cited a detailed video protocol which is fully open access where further details on how the proteins are prepared for this assay are provided.

4. *Worth noting that several studies have shown that RH5 is a target of acquired immunity and antibodies correlate with protection since these findings support RH5 as a vaccine candidate. Interestingly, it seems that the studies by both Richards et al 2013 and Osier et al 2014 report that antibodies to P113 were associated with protection from malaria. It would be good to mention this as it adds some further interest to P113 as an immune target and vaccine candidate*

This is an excellent suggestion. We have now included a sentence in the discussion which makes this point and references both of these studies.

“There are few studies on *P. falciparum* P113, but antibody responses against P113 have been shown to be associated with protection from malaria in study cohorts from both Africa [Osier 2014] and Papua New Guinea [Richards 2013]”.

References:

1. Baum J, *et al.* Reticulocyte-binding protein homologue 5 - an essential adhesin involved in invasion of human erythrocytes by *Plasmodium falciparum*. *Int J Parasitol* **39**, 371-380 (2009).
2. Gilson PR, *et al.* Identification and stoichiometry of glycosylphosphatidylinositol-anchored membrane proteins of the human malaria parasite *Plasmodium falciparum*. *Mol Cell Proteomics* **5**, 1286-1299 (2006).
3. Ord RL, *et al.* A malaria vaccine candidate based on an epitope of the *Plasmodium falciparum* RH5 protein. *Malar J* **13**, 326 (2014).
4. Reddy KS, Amlabu E, Pandey AK, Mitra P, Chauhan VS, Gaur D. Multiprotein complex between the GPI-anchored CyRPA with PfRH5 and PfRipr is crucial for *Plasmodium falciparum* erythrocyte invasion. *Proc Natl Acad Sci U S A* **112**, 1179-1184 (2015).
5. Chen L, *et al.* An EGF-like protein forms a complex with Pfrh5 and is required for invasion of human erythrocytes by *Plasmodium falciparum*. *PLoS Pathog* **7**, e1002199 (2011).
6. Elsworth B, *et al.* Proteomic analysis reveals novel proteins associated with the *Plasmodium* protein exporter PTEX and a loss of complex stability upon truncation of the core PTEX component, PTEX150. *Cellular microbiology*, (2016).
7. Jones ML, Kitson EL, Rayner JC. *Plasmodium falciparum* erythrocyte invasion: a conserved myosin associated complex. *Mol Biochem Parasitol* **147**, 74-84 (2006).
8. Maenaka K, *et al.* Killer cell immunoglobulin receptors and T cell receptors bind peptide-major histocompatibility complex class I with distinct thermodynamic and kinetic properties. *J Biol Chem* **274**, 28329-28334 (1999).
9. Zenonos ZA, *et al.* Basigin is a druggable target for host-oriented antimalarial interventions. *The Journal of experimental medicine* **212**, 1145-1151 (2015).
10. Douglas AD, *et al.* Neutralization of *Plasmodium falciparum* merozoites by antibodies against Pfrh5. *J Immunol* **192**, 245-258 (2014).

11. Silmon de Monerri NC, *et al.* Global identification of multiple substrates for Plasmodium falciparum SUB1, an essential malarial processing protease. *Infect Immun* **79**, 1086-1097 (2011).
12. Crosnier C, *et al.* Basigin is a receptor essential for erythrocyte invasion by Plasmodium falciparum. *Nature* **480**, 534-537 (2011).
13. Manske M, *et al.* Analysis of Plasmodium falciparum diversity in natural infections by deep sequencing. *Nature* **487**, 375-379 (2012).
14. Bustamante LY, *et al.* A full-length recombinant Plasmodium falciparum PfRH5 protein induces inhibitory antibodies that are effective across common PfRH5 genetic variants. *Vaccine* **31**, 373-379 (2013).

Reviewers' comments:

Reviewer #1 (Remarks to the Author):

Review of Resubmission "P113 is a merozoite surface protein that binds the N-terminus of Plasmodium falciparum RH5" by Dr. Wright and colleagues.

In the revised submission the authors have provided strong additional support for the interaction between P113 and Rh5 that addresses many of the concerns from the initial submission. The revised submission still does not address the *in vivo* interaction between the two proteins, which as the authors themselves explain at length will be very hard to validate definitively, given the transitory nature of the proposed interaction. Invasion imaging might help a bit (but its limitations are discussed and I accept as they were with CyRPA/Rh5/RIPR), protein immunoprecipitation in the context of invasion might also help (but is very challenging) and, following some form of conditional KO of P113 or its mutagenesis, it might be possible to show P113's necessity to invasion like that of the Rh5 complex (outside the remit of this paper and not in and of itself proof either). Thus, admittedly, none of these are either easy or in their own rights proof (P113 may be essential for many reasons). On the balance of the extended data presented in the revised submission I would, therefore, support publication. I think it is important that the work is out there and that others are encouraged to explore the function of the P113 protein. The structural work with Prof. Higgins will be very informative if forthcoming.

One correction I would favor is that I do not find the imaging data consistent with plasma membrane localization - instead very close inspection of the IFA with anti-P113 supports a broadly peripheral localization. This is not inconsistent with plasma membrane but it could equally be many other localizations in the parasite cell (anything non-nuclear/ER/IMC or PV can give such a signal!). EM or super-resolved imaging might help add clarity (i.e. define membrane) but is an undertaking in and of itself. I would therefore suggest changing the wording to either "not inconsistent with plasma membrane localization" or "consistent with a peripheral localization" but I do not agree that the data support plasma membrane localization specifically.

Reviewer #2 (Remarks to the Author):

The most important contribution of this work is the identification of P113 as a new member of the RH5 complex. I appreciate the Authors' careful and extensive improvements in this revision. However I still have two comments which are critical to draw this conclusion.

Comment#1

As shown in Figure 2c in the revised manuscript, the Authors demonstrated that recombinant RH5Nt interacts with full length native P113. The reason why the Authors used this approach is because anti-RH5Nt antibody inhibited the binding between RH5 and P113. To confirm this, I strongly recommend the Authors to test the reverse experiment as follows:

- 1) Immunoprecipitate P113 from the parasite extracts (by NP40 or Triton)
- 2) Western blot with anti-RH5Nt and anti-RH5Ct antibodies.

I am sure this is still important to try because this experiment is able to prove the natural RH5-P113 complex formation.

Comment#2

IFA images in the Supplementary Figure 1 is not acceptable, because the localization of P113 is very important in this manuscript.

Please check the following points for the improvement.

- 1) This parasite specimen looks not fully matured (very small number of merozoites)
- 2) DIC is out of focus
- 3) Anti-P113 staining image has a very high background signal, such as outside of the schizont is stained (right bottom area), cytoplasm of merozoites also stained. Please optimize the antibody concentration.
- 4) DAPI signal is too strong.
- 5) MTIP signal is on the inner membrane complex, so that MTIP signal is lacking at the apical end of the merozoite (should not be entire plasma membrane-like staining).

Moreover, the schizont IFA is not enough because the Authors could not distinguish the P113 signal either from parasitophorous vacuole or from merozoite surface.

So, IFA of free merozoite is essential for this work. Please set up the following two IFA conditions.

- 1) Fixed free merozoites with Triton permeabilization stained with both anti-P113 and MTIP (both signal will be on the surface of merozoites)
- 2) 1) Fixed free merozoites without Triton permeabilization stained with both anti-P113 and MTIP (P113 signal alone will be on the surface of merozoites)

If above results are obtained, the Authors conclusion will be supported.

Reviewer #3 (Remarks to the Author):

The authors have provided some valuable additional data in the revised manuscript that helps support their conclusions. In addition they have made some modifications to the paper to make it clearer and have provided a thoughtful and considered response to the various issues raised in their response letter.

While clear demonstration of the interaction between native Pf113 and native RH5 by immunoprecipitation or pull-down approaches would be ideal, as well co-localisation of these proteins on the merozoite surface, they do highlight the challenges in achieving this due to the timing of these interactions and the current technical constraints. However, they have now provided new data showing pull-down of P113 by RH5Nt. Based on this I am inclined to accept their conclusions are valid based on their careful and extensive studies using recombinant proteins and the new data provided.

A further issue is providing evidence that P113 is located on the merozoite surface membrane. They now provide IF images showing that anti-P113 labels the merozoite membrane in schizonts. This is supportive but not very conclusive. It would be ideal if they could show surface labelling of unfixed merozoites free of the schizont. I don't want to unnecessarily delay the publication of this quality work, but I feel that establishing this is quite important to the conclusions being drawn, and there does not appear to be published data clearly confirming P113 localisation on the merozoite surface.

As a minor point, I think the model they present could be clearer (Fig 7). If they are making revisions to the manuscript then I recommend they improve this figure

Point-by-point response to referees' comments on revised manuscript.

NCOMMS-16-00495A

Galaway *et al.* "P113 is a merozoite surface protein that binds the N-terminus of *Plasmodium falciparum* RH5."

Please find our point-by-point response to the referees' comments below.

Reviewer #1

In the revised submission the authors have provided strong additional support for the interaction between P113 and Rh5 that addresses many of the concerns from the initial submission. The revised submission still does not address the in vivo interaction between the two proteins, which as the authors themselves explain at length will be very hard to validate definitively, given the transitory nature of the proposed interaction. Invasion imaging might help a bit (but its limitations are discussed and I accept as they were with CyRPA/Rh5/RIPR), protein immunoprecipitation in the context of invasion might also help (but is very challenging) and, following some form of conditional KO of P113 or its mutagenesis, it might be possible to show P113's necessity to invasion like that of the Rh5 complex (outside the remit of this paper and not in and of itself proof either). Thus, admittedly, none of these are either easy or in their own rights proof (P113 may be essential for many reasons).

On the balance of the extended data presented in the revised submission I would, therefore, support publication. I think it is important that the work is out there and that others are encouraged to explore the function of the P113 protein. The structural work with Prof. Higgins will be very informative if forthcoming.

One correction I would favor is that I do not find the imaging data consistent with plasma membrane localization - instead very close inspection of the IFA with anti-P113 supports a broadly peripheral localization. This is not inconsistent with plasma membrane but it could equally be many other localizations in the parasite cell (anything non-nuclear/ER/IMC or PV can give such a signal!). EM or super-resolved imaging might help add clarity (i.e. define membrane) but is an undertaking in and of itself. I would therefore suggest changing the wording to either "not inconsistent with plasma membrane localization" or "consistent with a peripheral localization" but I do not agree that the data support plasma membrane localization specifically.

Please see results described in response to the points raised by referee 2 below which show - using co-staining with antibodies recognising the established merozoites surface markers MSP1 and MSP9 - P113 expression in both early and late-stage schizonts as well as on the surface of free merozoites. We have amended the wording as appropriate and replaced the ambiguous staining shown in Supplementary Figure 1 with a more comprehensive staining description of P113 which is included as a separate panel (panel d) of a main manuscript figure – Figure 2d, and further supporting data in a revised Supplementary Figure 1; we have also included additional staining in the rebuttal below.

Reviewer #2

The most important contribution of this work is the identification of P113 as a new member of the RH5 complex. I appreciate the Authors' careful and extensive improvements in this revision. However I still have two comments which are critical to draw this conclusion.

Comment#1

As shown in Figure 2c in the revised manuscript, the Authors demonstrated that recombinant RH5Nt interacts with full length native P113. The reason why the Authors used this approach is because anti-RH5Nt antibody inhibited the binding between RH5 and P113. To confirm this, I strongly recommend the Authors to test the reverse experiment as follows:

1) Immunoprecipitate P113 from the parasite extracts (by NP40 or Triton)

2) Western blot with anti-RH5Nt and anti-RH5Ct antibodies.

I am sure this is still important to try because this experiment is able to prove the natural RH5-P113 complex formation.

We agree with the referee that this would provide additional support for the interaction and we did consider performing this experiment in our first revision but we did not do so for essentially the same reasons the referee mentions: we were concerned that our polyclonal anti-P113 antibodies (which are raised against the entire P113 ectodomain) would disrupt the P113-RH5 interaction. Despite these concerns, we decided to try the experiment. Before attempting to co-purify RH5Nt and P113 using anti-P113 antibodies, it was important to establish whether the antibodies raised against P113 did indeed prevent or interfere with the binding of RH5 since the high affinity antibodies would almost certainly outcompete the P113-RH5 interaction. As anticipated, we could show that the anti-P113 antibodies prevented the binding of RH5 using our AVEIXIS assay, demonstrating that these antibodies would not be suitable for this experiment (Figure R1).

Figure R1. Polyclonal antibodies to the entire ectodomain of P113 block the binding of RH5Nt to P113. The RH5Nt-P113 interaction was detected using the AVEIXIS assay with P113 immobilised as the biotinylated “bait” and RH5Nt presented as the pentameric enzyme-tagged “prey”. Binding is indicated as the capture of the enzyme-tagged prey protein using the hydrolysis of an enzyme substrate to produce a product that absorbs at 485nm. The indicated concentrations of serially diluted protein G-purified polyclonal antibodies were incubated prior to addition of the prey to the bait. Antibodies to P113, the N-terminus of RH5 (RH5Nt) and to a lesser extent full-length RH5 (RH5FL) prevent the interaction, whereas antibodies to the C-terminus of RH5 (RH5Ct) do not block the interaction as expected. Data points represent means \pm 95% CI; $n = 3$.

To try and circumvent this problem, we attempted to remove those antibodies with epitopes in the RH5 binding region of P113 by pre-adsorbing the polyclonal antiserum with a recombinant P113 protein consisting of just the N-terminal region, representing the minimal RH5-binding region of P113. We re-tested these pre-adsorbed antibodies in our RH5-P113 AVEIXIS binding assay and could show that they no longer inhibited the RH5-P113 interaction (Figure R2).

Figure R2. Inhibition of the RH5-P113 interaction by anti-P113 polyclonal antibodies can be relieved by pre-adsorption with a recombinant protein fragment of P113 corresponding to the RH5 binding region. The RH5FL-P113 interaction was detected using the AVEIXIS assay with P113 immobilised as the biotinylated “bait” and RH5FL presented as the pentameric enzyme-tagged “prey”. Binding is indicated as the capture of the enzyme-tagged prey protein using the hydrolysis of an enzyme substrate to produce a product that absorbs at 485nm. Serial dilutions of antibodies to P113 (filled triangles) or those that were first preadsorbed against a protein corresponding to the RH5-binding region of P113 (open triangles) were tested for their ability to block the RH5-P113 interaction. While P113 antibodies blocked the interaction at higher concentrations, anti-P113 antibodies that were preadsorbed with a P113 fragment corresponding to the RH5 binding site did not. Data points represent means \pm 95% CI; $n = 3$.

We therefore attempted to biochemically purify P113 from late schizont extracts using anti-P113 antibodies that had been preadsorbed with the RH5 binding region of P113. Unfortunately, the preadsorption of the antibodies with epitopes in the RH5 binding region consistently and significantly reduced the amount of P113 that could be immunoprecipitated (Figure R3). We attempted to detect RH5 in these immunoprecipitates by Western blotting but even with extended exposure times, we were not able to detect any RH5 (data not shown). We also attempted the same experiment without preadsorbing the anti-P113 with RH5-binding region, and while the amount of immunoprecipitated P113 increased, perhaps because the anti-P113 antibodies blocked or outcompeted the P113-RH5 interaction (Figure R1), we again did not detect any RH5.

While we understand the referee's concern about the use of recombinant rather than native proteins to identify and validate interactions, we would like to make the point that this recombinant protein approach has been very successful in identifying extracellular interactions that have demonstrable *in vivo* relevance and which, because of the biochemical difficulties of working with membrane-embedded receptor proteins, would almost certainly not have been discovered using native protein based approaches. The main motivation behind developing the AVEIXIS approach was to address the technical difficulties of detecting the often low affinity interactions between membrane proteins which are very hard to solubilise in detergents which retain their native structure – a point made in detail here: Wright GJ. "Signal initiation in biological systems: the properties and detection of transient extracellular protein interactions." *Mol Biosyst.* 2009 Dec;5(12):1405-12. PMID: 19593473. Importantly, this approach requires that all of the proteins are expressed in mammalian cells which (for extracellular proteins) increases the chances that the structurally critical disulphide bonds are faithfully added; by contrast, extracellular proteins expressed using the more widely used bacterial systems, have a much lower chance of producing a correctly folded protein.

Examples of where we have validated the *in vivo* relevance of interactions discovered using recombinant proteins by AVEIXIS using gene-deficient animal models are: Bianchi E, Doe B, Goulding D, Wright GJ. "Juno is the egg Izumo receptor and is essential for mammalian fertilization." *Nature* 2014 Apr 24;508(7497):483-7. PMID: 24739963 and Powell GT, Wright GJ. "Jamb and jamc are essential for vertebrate myocyte fusion." *PLoS Biol.* 2011 PMID: 22180726. We'd also like to point out that the basigin-RH5 interaction which is now functionally well validated by the wider malaria community (including a co-crystal structure of recombinantly-expressed proteins) was discovered using recombinant proteins by AVEIXIS; and yet, presumably due to the technical difficulties in working with membrane-embedded receptor proteins no-one - to our knowledge - has yet reported an interaction between the native RH5 and basigin proteins. In short, we believe that when used appropriately recombinant proteins are an excellent approach to identify novel interactions that are functionally relevant *in vivo*, especially when detecting low affinity extracellular interactions between receptor proteins which are often biochemically intractable in their native membrane-embedded form.

Commet#2

IFA images in the Supplementary Figure 1 is not acceptable, because the localization of P113 is very important in this manuscript.

Please check the following points for the improvement.

- 1) This parasite specimen looks not fully matured (very small number of merozoites)*
- 2) DIC is out of focus*
- 3) Anti-P113 staining image has a very high background signal, such as outside of the schizont is stained (right bottom area), cytoplasm of merozoites also stained. Please optimize the antibody concentration.*
- 4) DAPI signal is too strong.*

5) MTIP signal is on the inner membrane complex, so that MTIP signal is lacking at the apical end of the merozoite (should not be entire plasma membrane-like staining).

Moreover, the schizont IFA is not enough because the Authors could not distinguish the P113 signal either from parasitophorous vacuole or from merozoite surface.

So, IFA of free merozoite is essential for this work. Please set up the following two IFA conditions.

1) Fixed free merozoites with Triton permeabilization stained with both anti-P113 and MTIP (both signal will be on the surface of merozoites)

2) Fixed free merozoites without Triton permeabilization stained with both anti-P113 and MTIP (P113 signal alone will be on the surface of merozoites)

If above results are obtained, the Authors conclusion will be supported.

We accept the points made by the referee and agree that these are important experiments that would lend strong support to our conclusions. We have now performed a more detailed characterisation of P113 expression at the protein level in the blood stages using antibodies against P113 together with co-staining with the well-established merozoite surface markers, MSP1 and MSP9. These data show high levels of P113 expression that co-localises with MSP1 and MSP9 in the late stage (segmenting) schizont and free merozoites. These important data support our conclusions and so have been added to the main manuscript as co-staining with anti-MSP9 in the schizont and free merozoites in a modified Figure 2d, and with anti-MSP1, including early and late schizonts as a revised Supplementary Figure 1.

We include here in the rebuttal a representative wide-field view of a P113/MSP9 co-staining experiment from a recently ruptured schizont which shows consistent staining on the surface of several free merozoites (Figure R4).

Figure R4. P113 is expressed in schizonts and on the merozoite surface. Highly synchronised *P. falciparum* blood stage cultures were stained with anti-P113 (green) and anti-MSP9 (red) before being counter-stained with DAPI (blue) to show nucleic acid and imaged by confocal microscopy. Scale bars represent 7.5µm.

Finally, since the submission of the revision of this manuscript other researchers have independently performed similar experiments and concluded that P113 is localised on the surface of the merozoite. See: “Immunoglobulin response to the low polymorphic Pf113 antigen in children from Lastoursville, South-East of Gabon” *Acta Tropica* 2016 vol. 163 p149 – we reference this paper in the appropriate place in our revised manuscript.

Reviewer #3

The authors have provided some valuable additional data in the revised manuscript that helps support their conclusions. In addition they have made some modifications to the paper to make it clearer and have provided a thoughtful and considered response to the various issues raised in their response letter.

While clear demonstration of the interaction between native Pf113 and native RH5 by immunoprecipitation or pull-down approaches would be ideal, as well co-localisation of these proteins on the merozoite surface, they do highlight the challenges in achieving this due to the timing of these interactions and the current technical constraints. However, they have now provided new data showing pull-down of P113 by RH5Nt. Based on this I am inclined to accept their conclusions are valid based on their careful and extensive studies using recombinant proteins and the new data provided.

A further issue is providing evidence that P113 is located on the merozoite surface membrane. They now provide IF images showing that anti-P113 labels the merozoite membrane in schizonts. This is supportive but not very conclusive. It would be ideal if they could show surface labelling of unfixed merozoites free of the schizont. I don't want to unnecessarily delay the publication of this quality work, but I feel that establishing this is quite important to the conclusions being drawn, and there does not appear to be published data clearly confirming P113 localisation on the merozoite surface.

Please see results described in response to the points raised by referee 2 above which show P113 staining on the surface of free merozoites.

As a minor point, I think the model they present could be clearer (Fig 7). If they are making revisions to the manuscript then I recommend they improve this figure.

We have modified the Figure 7 legend which we hope improves its legibility, although we were mindful not to be redundant with the detailed description of the model in the discussion.

REVIEWERS' COMMENTS:

Reviewer #2 (Remarks to the Author):

I appreciate the Authors' careful and extensive improvements again in this re-revision. Quality of these IFA pictures in Figure 2d are significantly improved (both mature schizonts & free merozoites). However, one important point needs to be revised.

The Authors fixed the parasite specimen in ice-cold methanol, then permeabilized in PBS/Triton X100. But permeabilization make this beautiful data inconclusive. One of the most important points of this work is to confirm the surface expression of Pf113. So, please re-design this IFA experiments with non-permeabilized parasite specimen (both mature schizonts & free merozoites) then stain them using anti-Pf113 antibody with anti-MTIP. If Pf113 is surface expression, the Authors will be able to stain Pf113 on free merozoites but not MTIP. If both Pf113 & MTIP are visualized, parasite plasma membrane (pellicle) is artificially disrupted.

Reviewer #3 (Remarks to the Author):

The authors have addressed my request for further imaging studies to support the conclusion that P113 is on the merozoite surface.

Point-by-point response to referees' comments on revised manuscript.

NCOMMS-16-00495B

Galaway *et al.* "P113 is a merozoite surface protein that binds the N-terminus of *Plasmodium falciparum* RH5."

Please find our point-by-point response to the referees' comments below.

Reviewer #1

No further comments were received.

Reviewer #2

I appreciate the Authors' careful and extensive improvements again in this re-revision. Quality of these IFA pictures in Figure 2d are significantly improved (both mature schizonts & free merozoites). However, one important point needs to be revised.

The Authors fixed the parasite specimen in ice-cold methanol, then permeabilized in PBS/Triton X100. But permeabilization make this beautiful data inconclusive. One of the most important points of this work is to confirm the surface expression of Pf113. So, please re-design this IFA experiments with non-permeabilized parasite specimen (both mature schizonts & free merozoites) then stain them using anti-Pf113 antibody with anti-MTIP. If Pf113 is surface expression, the Authors will be able to stain Pf113 on free merozoites but not MTIP. If both Pf113 & MTIP are visualized, parasite plasma membrane (pellicle) is artificially disrupted.

We have revisited this question by removing the PBS/Triton X-100 permeabilisation step, and as the referee suggests performed co-staining with antibodies against P113 and MTIP (Figure R1).

Figure R1. Antibodies to P113, but not a marker of the inner membrane complex (MTIP), stain the surface of unpermeabilised merozoites. Fixed merozoites were stained either with (+) or without (-) a membrane permeabilisation step using the non-ionic detergent Triton X-100 and then co-stained with the DAPI nuclear dye. Scale bar represents 3 μ m.

These data lend additional support localising P113 to the merozoites surface and so we have included them as an additional panel in Supplementary Figure 1a.

Reviewer #3

The authors have addressed my request for further imaging studies to support the conclusion that P113 is on the merozoite surface.

Many thanks, and please see response to referee 2 above.